# Photosymbiont associations persisted in planktic foraminifera during early Eocene hyperthermals at Shatsky Rise (Pacific Ocean)

Catherine V. Davis [1,2]*, Jack O. Shaw [2¤a], Simon D'haenens [2¤b], Ellen Thomas [2,3], Pincelli M. Hull[2,4]

**1** Department of Marine, Earth, and Atmospheric Sciences, North Carolina State University, Raleigh, NC, United States of America, **2** Department of Earth and Planetary Sciences, Yale University, New Haven, CT, United States of America, **3** Department of Earth and Environmental Sciences, Wesleyan University, Middletown, CT, United States of America, **4** Peabody Museum of Natural History, Yale University, New Haven, CT, United States of America

¤a Current address: Santa Fe Institute, Santa Fe, NM, United States of America
¤b Current address: Data Science Institute & Research Coordination Office, Hasselt University, Hasselt, Belgium
* catherinedavis@ncsu.edu

**Data Availability Statement:** All relevant data are within the paper and its Supporting Information files.

## Abstract

Understanding the sensitivity of species-level responses to long-term warming will become increasingly important as we look towards a warmer future. Here, we examine photosymbiont associations in planktic foraminifera at Shatsky Rise (ODP Site 1209, Pacific Ocean) across periods of global warming of differing magnitude and duration. We compare published data from the Paleocene-Eocene Thermal Maximum (PETM; ~55.9 Ma) with data from the less intense Eocene Thermal Maximum 2 (ETM2; ~54.0 Ma), and H2 events (~53.9 Ma). We use a positive relationship between test size and carbon isotope value (size-$\delta^{13}$C) in foraminifera shells as a proxy for photosymbiosis in *Morozovella subbotinae* and *Acarinina soldadoensis*, and find no change in photosymbiont associations during the less intense warming events, in contrast with PETM records indicating a shift in symbiosis in *A. soldadoensis* (but not *M. subbotinae*). Declines in abundance and differing preservation potential of the asymbiotic species *Subbotina roesnaesensis* along with sediment mixing likely account for diminished differences in $\delta^{13}$C between symbiotic and asymbiotic species from the PETM and ETM2. We therefore conclude that photosymbiont associations were maintained in both *A. soldadoensis* and *M. subbotinae* across ETM2 and H2. Our findings support one or both of the hypotheses that 1) changing symbiotic associations in response to warming during the PETM allowed *A. soldadoensis* and perhaps other acarininids to thrive through subsequent hyperthermals or 2) some critical environmental threshold value was not reached in these less intense hyperthermals.

## Introduction

Rapid warming in marine environments can be catastrophic: the largest mass extinction of life on Earth, the Permian-Triassic Extinction, has been linked to warming caused by massive

**Funding:** ET recognizes funding by National Science Foundation (NSF) OCE 1536611. PMH, SD, and JOS recognize funding by NSF OCE 1536604 and a Sloan Research Fellowship.

**Competing interests:** The authors have declared that no competing interests exist.

release of greenhouse gasses (e.g., [1, 2]). In contrast, more modest warming events of the Cenozoic had mixed effects on species and ecosystems (e.g., [3–5]), with some clades, especially benthic foraminifera, experiencing mass extinctions while others exhibited adaptive responses, including changes in range, abundance, and body size [5–7]. In the pelagic ocean, changes in the composition of calcareous phytoplankton at the community level scale with the magnitude of warming during early Eocene hyperthermals at a Pacific site [8]. However, the relation has been shown for one group of phytoplankton only, and the adaptive response and survival of individual species or genera is unclear. As we enter a period of rapid warming, we must understand how individual species respond to varying degrees of warming (and associated environmental changes) in order to craft effective policies.

In the modern ocean, some photosymbiotic taxa, including corals and reef-dwelling large benthic foraminifera, expel or reduce their photosymbiont load ('bleach'), a response which appears detrimental to growth and survival, during short-term fluctuations in photoexposure and/or increases in marine temperatures of just ~1–3° C above local maxima [9–13]. A similar response has been proposed for some species of extinct planktic foraminifera, calcifying marine protists that provide an extensive fossil record of past warming. Their shell chemistry records near-surface environments and changes in biology and ecology, including symbiosis. It has been proposed that there have been relatively long (geologically evident) periods during which formerly symbiotic species of foraminifera either lost or reduced their photosymbiotic relationships [14–17].

Photosymbiosis in modern and fossil foraminifera can be identified primarily by (1) a positive relationship between shell size and $\delta^{13}C$ (size-$\delta^{13}C$), and secondarily by (2) $\delta^{13}C$ enrichment of the whole shell with respect to coexisting asymbiotic taxa [18–23]. Decreased light levels, or a reduction in photosynthesis, result in decreased size-$\delta^{13}C$ and $\delta^{13}C$ enrichment in modern foraminifera [19]. In addition, different symbiont types (e.g., dinoflagellate versus chrysophyte or pelagophyte) and arrangements produce different $\delta^{13}C$ signatures in both living and extinct species (e.g., [21, 22, 24–26]). A loss, reduction, or change in photosymbiotic associations can thus be deduced based on changes in $\delta^{13}C$ values, primarily a decrease in the size-$\delta^{13}C$ gradient [14–16]. Both proxies can be used independent of global or local $\delta^{13}C$ values in specimens which lived at the same time and place, and therefore experienced the same macro-environmental $\delta^{13}C$ over their lifespan. Thus, $\delta^{13}C$ values within a time slice and between species sharing a common ambient carbon environment reflect the microenvironment of the holobiont, even during changes in the global carbon cycle such as those associated with hyperthermals.

To understand the long-term response of photosymbiotic taxa across warming events of varying severity, we look to three early Eocene hyperthermal events of different magnitude: the Paleocene-Eocene Thermal Maximum (PETM, ~55.9 Ma), and the double peaks of Eocene Thermal Maximum 2, sometimes called the H1 (ETM2, ~54.0 Ma), and H2 (~53.9 Ma) events (e.g., [27]). During the PETM, the most extreme of the Eocene hyperthermals, sea-surface temperatures (SST) rose by 4–8° C (4–5° C at Shatsky Rise [28, 29]), due to carbon emissions potentially greater than 10,000 Pg [30]. In contrast, ETM2 SST records show an increase of ~3° C at several sites, including Shatsky Rise [31, 32], with a warming of ~2° C during the smaller H2 event in the Southeast Atlantic [33].

Changes in photosymbiotic associations as deduced from $\delta^{13}C$ records have been described for several periods of warming, including the PETM [34, 35], the Early Eocene Climatic Optimum [16, 36, 37] and during the Middle Eocene Climatic Optimum (~40 Ma) [15]. A loss of symbionts has also been reported during periods when there was no warming, including following the Middle Eocene Climatic Optimum [14] and around the onset of Northern Hemisphere glaciation [17]. These changes have been interpreted as either analogous to 'bleaching'

in modern benthic foraminifera and corals [14–16], or as a reflection of a more moderate, or adaptive response to warming, such as a change in symbiont type, density, or arrangement [16, 25]. However, it is difficult to clearly distinguish changes in photosymbiont status across past warming events due to multiple confounding factors. These factors include size dependent sediment mixing, with preferential upward mixing of smaller individuals into younger sediments artificially reducing size-$\delta^{13}$C during isotopic excursions [38, 39]. Additionally, changes in light levels and/or foraminiferal depth habitats could have decreased photosymbiont activity, thus reducing both the size-$\delta^{13}$C correlation and the differences in $\delta^{13}$C between symbiotic and asymbiotic taxa [35]. Finally, spatial and taxonomic heterogeneity in the response of planktic foraminifera to early Eocene warming [34, 35, 40] demonstrates that responses to environmental perturbations were at least in some cases regional and species-specific.

To evaluate whether rapid warming and/or associated changes in carbon cycling impacted species-specific symbiont associations consistently, we focus on three species at a single site over the course of multiple hyperthermals. Specifically, we compare three hyperthermal events of differing magnitude at ODP Site 1209 (Shatsky Rise, Pacific Ocean), where symbiont change across the PETM has been evaluated and identified in *Acarinina soldadoensis* [34]. We target the same species and assess symbiosis by size-$\delta^{13}$C and the difference in $\delta^{13}$C between symbiotic (*A. soldadoensis* and *Morozovella subbotinae*) and asymbiotic (*Subbotina roesnaesensis*, identified as *S. eocaena* by [34]) species in the subsequent ETM2 and H2 hyperthermals.

## Materials and methods

Sediment samples from ODP Leg 198 Site 1209 (2387 m water depth) were requested from the IODP Core Repository. Bulk samples were disaggregated with a sodium metaphosphate solution, washed over a 63 μm sieve, and dried at 40° C. Data from eleven intervals centered on ETM2 (n = 7) and H2 (n = 4) are compared with data from PETM intervals (n = 7) [34], with pre, peak, and post event intervals identified using the chronology of [41] (Figs 1 and 2). Dried sediments were sieved into six standard size fractions (150–180, 180–212, 212–250, 250–300, 300–355, and >355 μm), from which planktic foraminifera were picked, with size fractions referred to by their lower boundary. Planktic foraminifera from the species *A. soldadoensis*, *M. subbotinae*, and *S. roesnaesensis*, were selected following the taxonomy of [42] (S1 Fig).

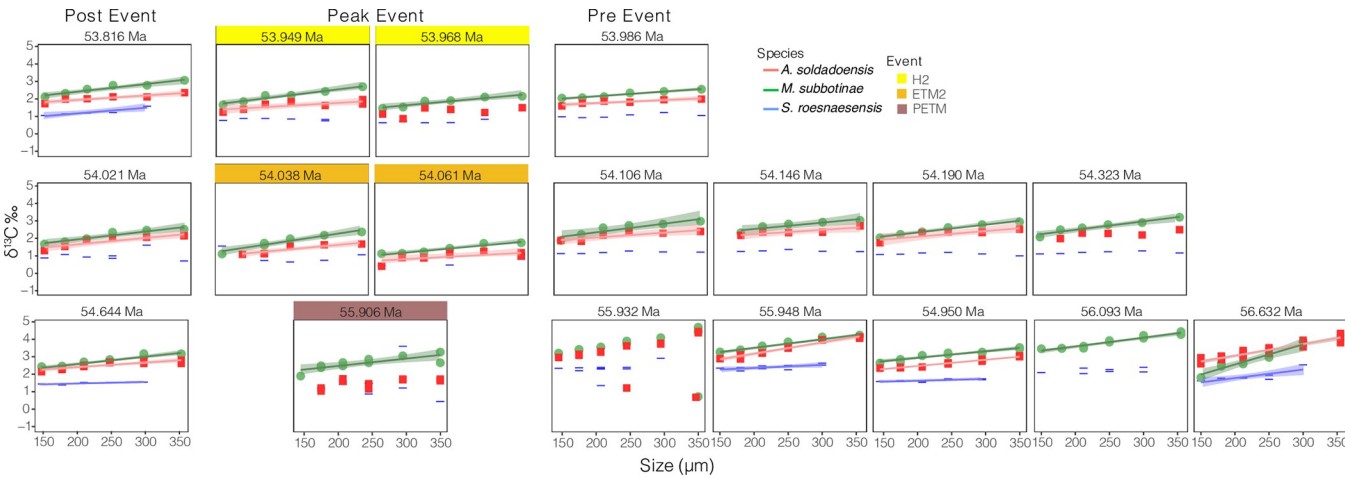

**Fig 1. $\delta^{13}$C relative to size class for each interval analyzed.** Intervals designated as 'peak' events are shaded in yellow, orange, and brown for the H2, ETM2, and PETM events. Intervals proceeding and following "peak" intervals represent those considered "pre" and "post" event samples, respectively. Significant (p-value < 0.05) linear fits are shown in each plot with a standard error envelope at the 95% confidence interval. PETM data is from [34].

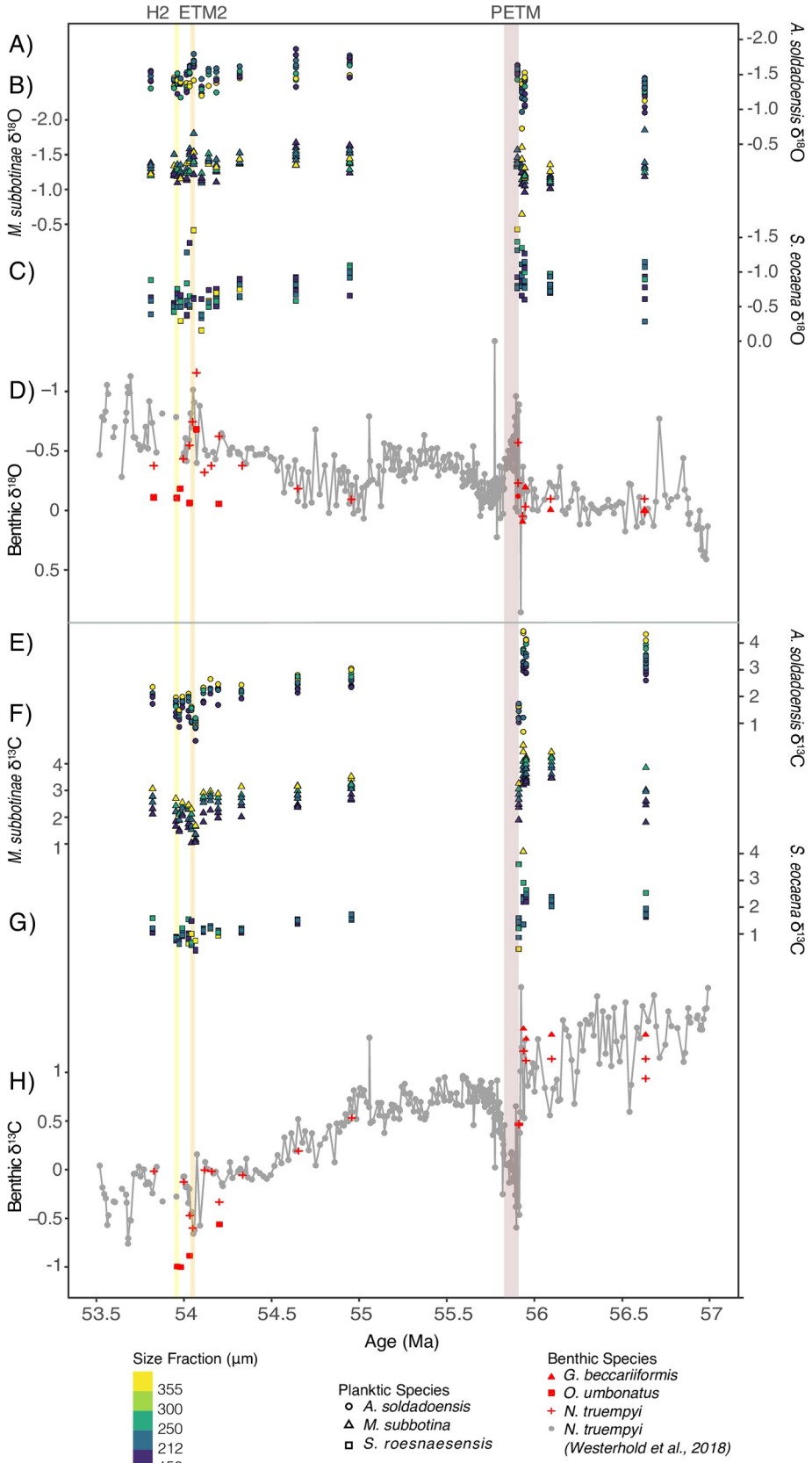

**Fig 2.** δ<sup>18</sup>O records at Site 1209 from A) *A. soldadoensis*, B) *M. subbotinae*, C) *S. roesnaesensis*, and D) benthic *Nuttallides truempyi*, along with δ<sup>13</sup>C records from E) *A. soldadoensis*, F) *M. subbotinae*, G) *S. roesnaesensis*, and H) benthic *N. truempyi*. Data from the period prior to 54.323 Ma is from [34]. Benthic foraminiferal records are from [41] (gray) and this study (red), with species denoted by symbol shape. Planktic foraminifera size fraction is denoted by color. The H2, ETM2, and PETM intervals are designated with yellow, orange, and brown shading respectively. All records follow the chronology of [41].

We aimed to pick at least thirty individuals of each target species from a size fraction, but the number of individuals picked varied from a maximum of ~60 in the finer fractions to a minimum of one specimen in the coarser fractions, because of the carbonate mass needed for isotopic analyses (S1 Data). Some size-specific sample fractions yielded no specimens. Between 5 and 10 individual benthic foraminifera of the species *Oridorsalis umbonatus* and *Nuttallides truempyi* were picked from the 150–212 μm size fraction. All foraminifera samples were sonicated for 5–10 s prior to analysis using a Thermo Finnigan MAT 253 coupled to a Kiel IV Carbonate Preparation Device in the Yale Analytical and Stable Isotope Center, with an external analytical precision (1σ) of 0.013‰ for δ<sup>13</sup>C and 0.018‰ for δ<sup>18</sup>O. A total of 199 size-fraction and species-specific samples of planktic foraminifera were analyzed for δ<sup>13</sup>C and δ<sup>18</sup>O, supplemented by data from the PETM [34] (S1 Data). Isotopic analyses are destructive, thus analyzed specimens are no longer available, but both data and images of all specimens are available in the Supplementary Data (S1, S2 Data).

Size-δ<sup>13</sup>C data for *A. soldadoensis*, *M. subbotinae*, and *S. roesnaesensis* were additionally compared with data across the same intervals from neighboring Shatsky Rise ODP Site 577 [19]. Due to both the degree of taxonomic reassessment since publication [19] and the relatively low temporal resolution of this study, we present this data at genus level. Size-δ<sup>13</sup>C values from [19] were analyzed from multiple sieve fractions (90–106, 106–125, 125–150, 150–180, 180–212, 212–250, 250–300, 300–355, and >355 μm) and represent data from several species within *Acarinina* (*A. soldadoensis*, *A. wilcoxensis*, *A. mckannai*, and *A. praeangulata*), *Morozovella* (*M. lensiformis*, *M. subbotinae*, *M. velascoensis*, *M. occlusa*, *M. acuta*, *M. parva*, *M. pusilla*, *M. finchi*, *M. angulata*, *M. conicotruncata*, and *M. uncinate*), and *Subbotina* (*S. hornibrooki*, *S. linaperta*, *S. triangularis*, *S. velascoensis*, *S. praecursoria*, and *S. pseudobulloides*) as identified by the authors (Fig 3).

Abundances were assessed using 500–700 individuals from a split of the >150 μm fraction in the same 11 samples from which isotopes were analyzed. Genus level identifications of *Acarinina*, *Morozovella*, and *Subbotina* were made, with results presented in terms of relative abundance. Test fragments were defined as shells missing multiple chambers, and they are reported relative to the total number of shells and fragments counted in a sample. Due to the use of the >150 μm sieve fraction, it is unlikely but not impossible that multiple fragments from a single test were counted, but small fragments would not have been captured.

## Results

### Size-δ<sup>13</sup>C isotope record over ETM2 and H2 at Site 1209

Planktic foraminifera from Site 1209 are frosty with moderate preservation throughout the sampled interval [43], as is usual for specimens from carbonate ooze, with some variation and generally better preservation in younger intervals. Frosty preservation increases planktic shell δ<sup>18</sup>O values due to recrystallization in relatively cold bottom waters [44, 45]. As a result, we make no attempt to interpret δ<sup>18</sup>O values here. However, frosty preservation has a relatively minor impact on foraminiferal δ<sup>13</sup>C [44, 45] and should not significantly impact size-δ<sup>13</sup>C gradients within a sample. Thus, we interpret the relative changes in δ<sup>13</sup>C values as providing meaningful information about the ecology and calcification environment of these shells.

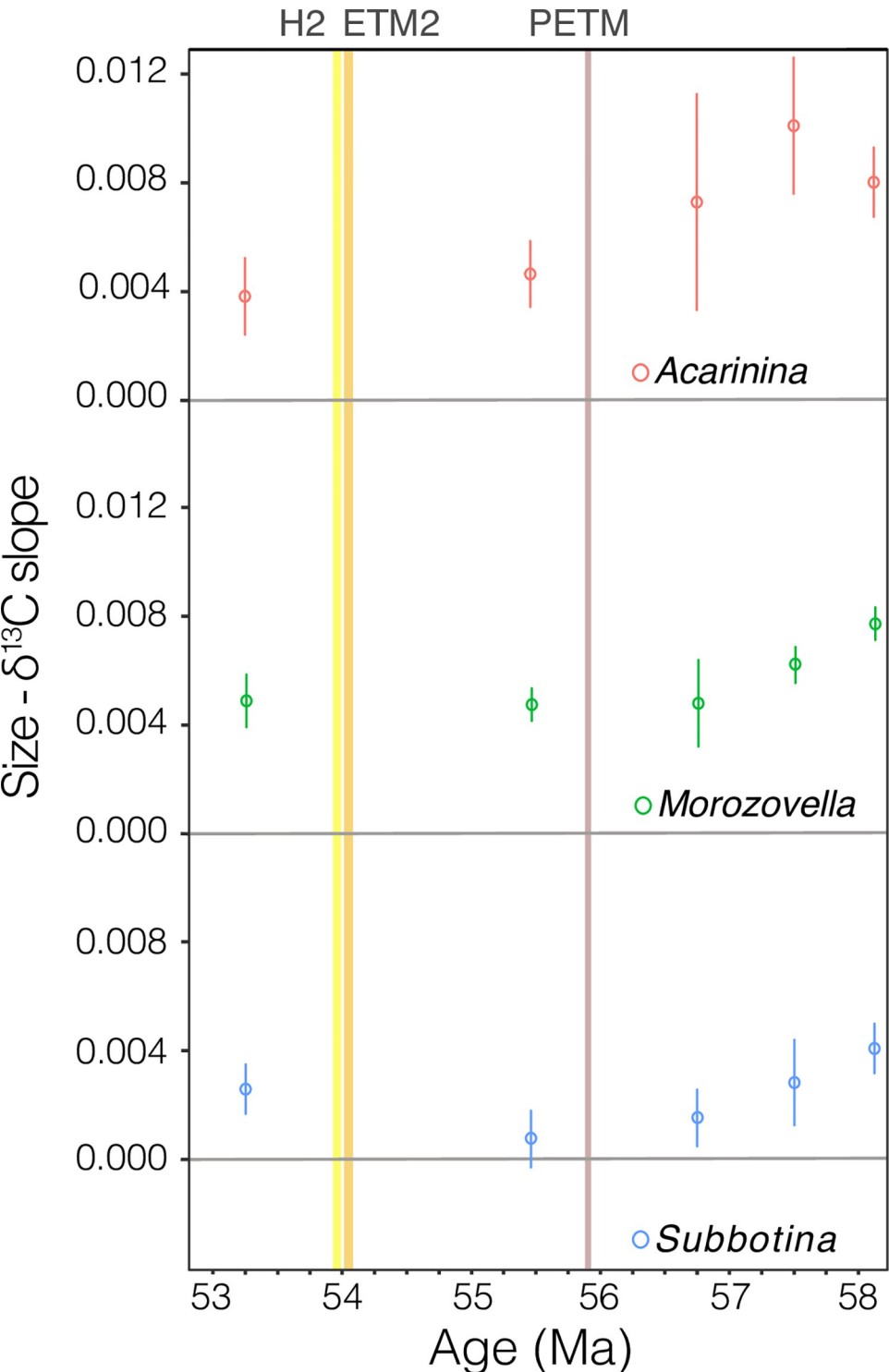

**Fig 3. The slope of δ¹³C relative to size at the genus level for *Acarinina* (red), *Morozovella* (green) and *Subbotina* (blue) species from ODP Site 577, after [19], rescaled to the age model of [41] and shown as open circles with error bars.** The H2, ETM2, and PETM intervals are designated with yellow, orange, and brown shading respectively.

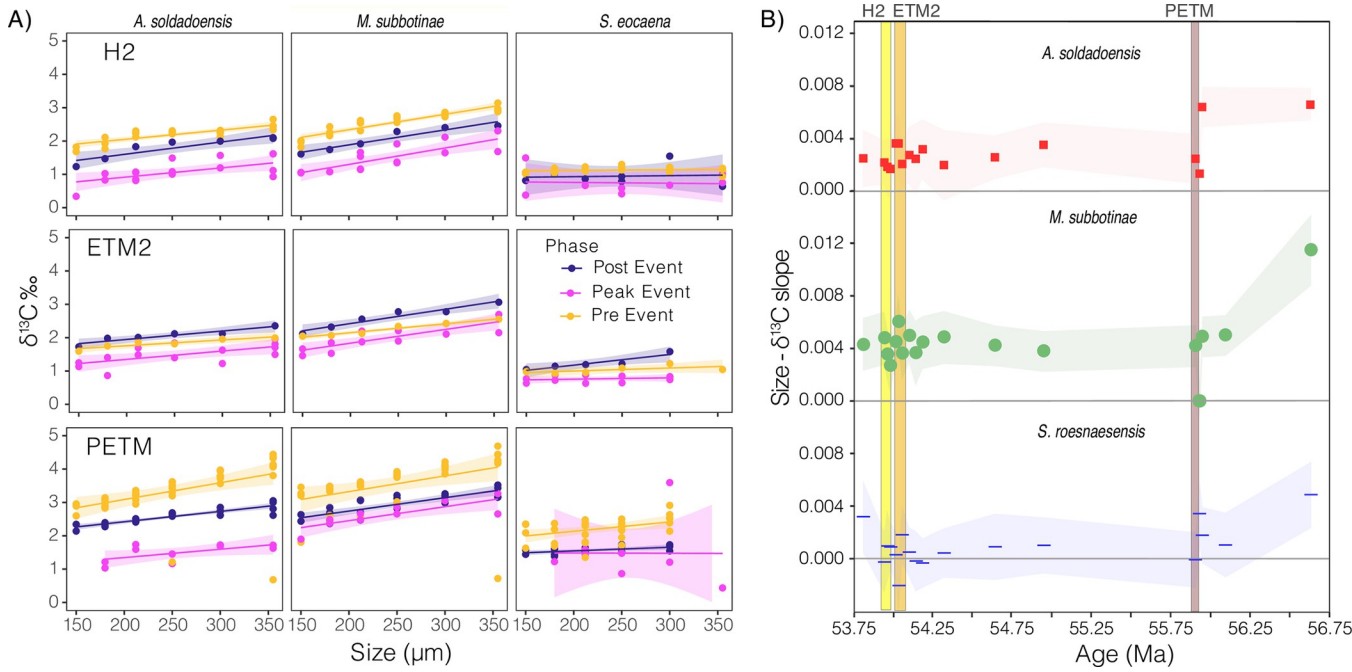

**Fig 4.** A) $\delta^{13}$C relative to size fraction at peak (pink) H2, ETM2, and PETM and pre and post event (yellow and purple respectively) intervals for *A. soldadoensis*, *M. subbotinae*, and *S. roesnaesensis*. Multiple time-slices are binned as "pre" "peak" and "post" event (Fig 1, S1 Fig & S1 Table). Linear fits for each relationship are shown with a standard error envelope. B) The slope of $\delta^{13}$C relative to size for *A. soldadoensis* (red squares), *M. subbotinae* (green circles), and *S. roesnaesensis* (blue lines) through time at Site 1209. The H2, ETM2, and PETM intervals are designated with yellow, orange, and brown shading respectively.

We find no significant size-$\delta^{13}$C decline in samples associated with peak ETM2 and H2 events in either symbiont-bearing species (i.e., linear regression slopes are within error), with positive, non-zero slopes retained for all samples (Fig 4). *Acarinina soldadoensis* and *M. subbotinae* show a slight increase in size-$\delta^{13}$C at the peak ETM2 relative to pre- or post-event levels, and a slight decrease in H2 (Figs 2 and 4; S2 Fig), but neither difference is statistically significant (Fig 4; S1 Table). Analysis of covariance (ANCOVA) tests indicate no significant interaction between shell size and event phase, as related to $\delta^{13}$C for *A. soldadoensis*, *M. subbotinae*, and *S. roesnaesensis* (S1 Table).

As a measure of the enrichment of $\delta^{13}$C in symbiotic species relative to asymbiotic species (independent of size-$\delta^{13}$C slope), we compared the average $\delta^{13}$C values of the smaller size fractions (mean of the 150 and 180 µm sieve fractions). In all size fractions, $\delta^{13}$C values of symbiotic *A. soldadoensis* and *M. subbotinae* are more similar to those of asymbiotic *S. roesnaesensis* during warming events than outside these events (compare smaller size fractions plotted in Fig 4). The difference in $\delta^{13}$C between *A. soldadoensis* and *S. roesnaesensis* ($\Delta\delta^{13}C_{Aca-Sub}$) collapses in both the PETM and ETM2. The $\Delta\delta^{13}C_{Mor-Sub}$ does not decrease notably in the peak PETM but shows changes of similar magnitudes to $\Delta\delta^{13}C_{Aca-Sub}$ in both peak ETM2 and peak H2 samples (Fig 5).

Acarininids and morozovellids are dominant across the ETM2 and H2 at Site 1209 (38–61% and 26–44% of the assemblage respectively), whereas subbotinids are present in all samples, but uncommon. The relative dominance between acarininids and morozovellids skews most strongly towards acarininids in two samples between the PETM and ETM2, and between ETM2 and H2. Acaranininds continue to dominate during both ETM2 and H2, whereas in the PETM morozovellids appeared to be the dominant genus (Fig 6). Variability in abundance is low in subbotinids, but the genus is rarer in hyperthermal events than in the intervening intervals, making up <8% of the assemblage in H2 (7.7% and 5.8%), <5% (3.5% and 4.9%) in

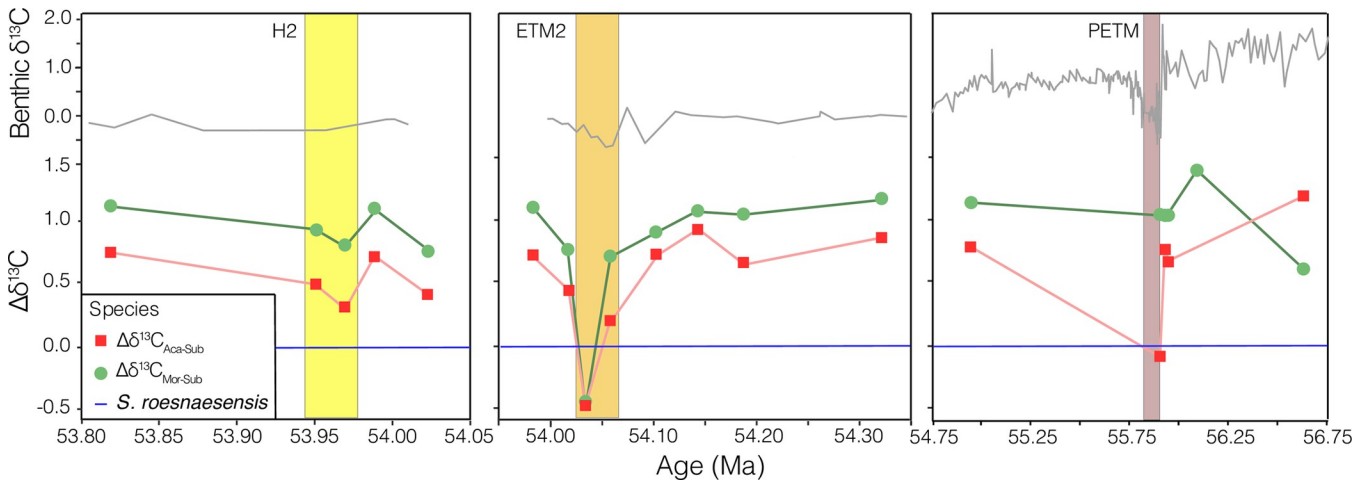

**Fig 5. The Δδ¹³C of *A. soldadoensis* (red) and *M. subbotinae* (green), relative to that of *S. roesnaesensis* (blue) through time.** Note different horizontal axes (time) across panels. The benthic δ¹³C record from [41] is shown in gray above, and the H2, ETM2, and PETM intervals are designated with yellow, orange, and brown shading respectively.

ETM2 and <10% for the duration of the PETM [43] (Fig 6). Fragmentation is relatively low, ranging between 5 and 12% throughout samples, without a consistent relationship to hyperthermals, in contrast to an increase in fragmentation over the PETM (Fig 5B; [43]).

## Discussion

### Symbiosis in *A. soldadoensis* and *M. subbotinae* maintained across ETM2 and H2

ETM2 and H2 are defined by global environmental change due to massive releases of greenhouse gases, as during the PETM, but on a lesser scale [31, 33, 46]. A shift in symbiont ecology

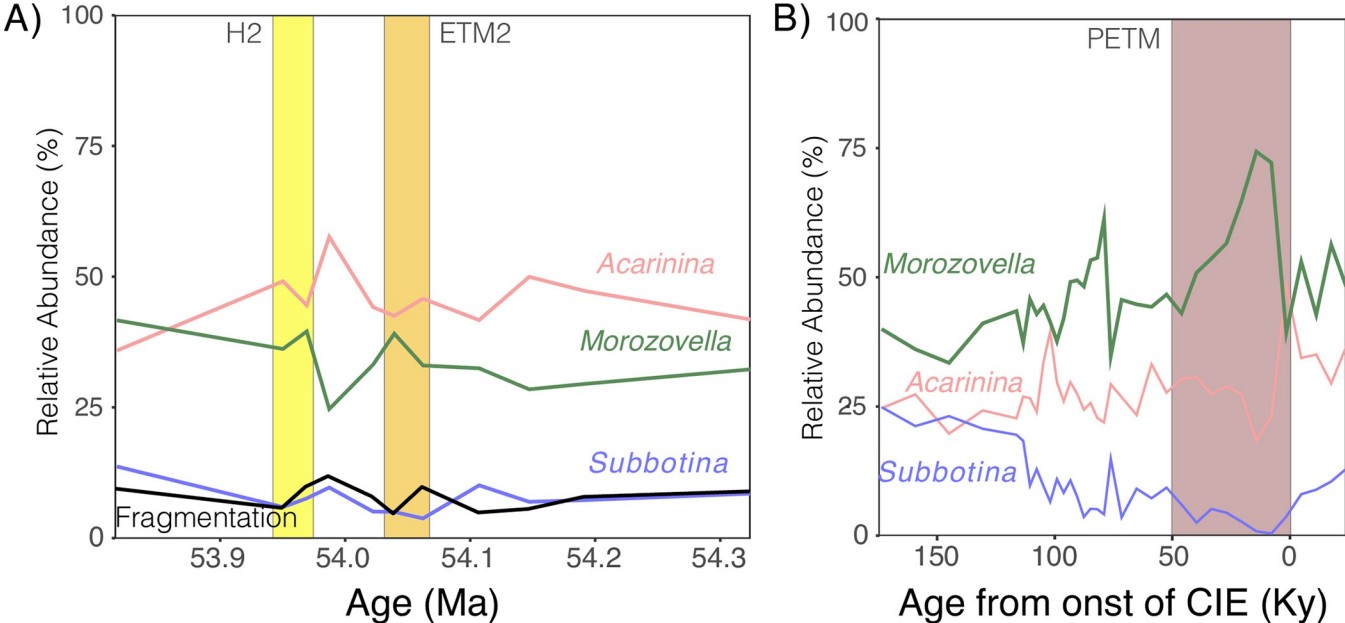

**Fig 6.** Relative abundance of planktic foraminifera genera (A) before, within, and after the H2 (yellow), ETM2 (orange) and (B) PETM (brown) events. The percent of fragmented tests is shown in black. PETM abundance data from with fragmentation data shown in (Fig 5B of that publication) [43].

(type, arrangement, or capacity), or other localized cause of size-$\delta^{13}$C collapse, in association with rapid climate change has been inferred across the PETM at Site 1209 in *A. soldadoensis* [34] and at the genus level for *Acarinina* in nearby Site 577 [19] (Fig 3; S3 Fig), but not in *M. subbotinae*. In contrast to the PETM, no decrease in size-$\delta^{13}$C was observed in *A. soldadoensis* or *M. subbotinae* across and following the ETM2 or H2 events at Site 1209 (Figs 1 and 2).

A reduction in maximum shell size has been associated with 'bleaching' during both the Middle Eocene Climatic Optimum and Early Eocene Climatic Optimum [16, 47]. Here, both *A. soldadoensis* and *M. subbotinae* are consistently present in the largest size fraction examined (>355 μm) in the ETM2 and H2 events at Shatsky Rise Site 1209 (Figs 1 and 2), and at Site 1210 planktic foraminifera were in fact larger during and following the PETM than before [48]. A sustained maximum size of >355 μm further supports the conclusion from size-$\delta^{13}$C data that symbiosis in these taxa was maintained across both less intense hyperthermals.

We considered the potential effects of size-dependent sediment mixing, a process which causes an apparent collapse of size-$\delta^{13}$C due to preferential post-depositional upward mixing of smaller shells in the sediment, as for example identified at Weddell Sea Site 690 [38, 39, 49]. Size-dependent sediment mixing may lead to a decrease or even an inversion of size-$\delta^{13}$C gradients across events with abrupt and strong CIEs, if pre or post event (higher $\delta^{13}$C) smaller shells are preferentially included in samples with larger peak event (lower $\delta^{13}$C) shells. Mixing may have contributed to the lack of a significant slope in three samples of *A. soldadoensis* (55.906, 54.323, and 53.968 Ma, Fig 1), two of which are associated with relatively low abundances of *Acaranina* compared to earlier periods. However, there is no deviation from a positive slope in either *A. soldadoensis* or *M. subbotinae* (Fig 4B). Moreover, when "peak" samples from all three hyperthermals are considered together (see methods), a significant positive gradient is maintained in *A. soldadoensis* and *M. subbotinae*, consistent with the maintenance of photosymbiosis in these species [19–22, 42] (Fig 4A).

## Rarity or absence of *Subbotina* in hyperthermals

At Site 1209, the PETM, ETM2, and H2 are all marked by a decrease in the relative abundance of subbotinids [43, 50]. Subbotinids were absent during the peak PETM [50] and make up << 10% of the foraminiferal assemblage for the duration of the PETM (Fig 6). Similarly, subbotinids became less abundant in ETM2 and H2 intervals (Fig 6). This decrease in subbotinid abundance in hyperthermals has been observed at multiple PETM sites from the high latitude Site 690 [34, 51], to relatively low-latitude Sites 1209 [43, 50], 401 [34, 52], and the Forada section [43, 50, 53], as well as tropical Site 865 [54]. Subbotinids also decreased in abundance at Walvis Ridge Site 1263 during the ETM2 and other hyperthermal events (ETM3, J), but not at Site 1051 [33]. Decreasing subbotinid abundance was also observed during the EECO at Pacific Site 577 and the Tethyan Possagno section [16, 36, 37], but they show little change in abundance at more eutrophic Northwest Atlantic Site 1051 [16, 36].

The rarity of subbotinids during peak hyperthermal events increases the likelihood for inclusion of upward-mixed, pre-event *Subbotina*, with the greatest effect to be expected in intervals with the lowest abundance of subbotinids (PETM followed by ETM2). Put otherwise, while mixing impacts all foraminifera, if one group (subbotinids, here) becomes increasingly rare across an abrupt boundary, the ratio of reworked to contemporaneous shells will increase across the event boundary, and sampling of that species will be skewed towards reworked material. Thus, the isotopic signature of shells from a species experiencing a decline in abundance will be more susceptible to mixing artifacts than a species that has remained abundant. The applicability of this hypothesis to early Eocene hyperthermal at Site 1209 is supported by the relative dampening of the $\delta^{13}$C excursion, especially in smaller shells, recorded in *S.*

*roesnaesensis* as compared to *A. soldadoensis*, *M. subbotinae*, or benthic foraminifera across the three events (Fig 1; S2 Fig). We note that one alternate hypothesis is an increase in depth habitat by both *A. soldadoensis* and *M. subbotinae*, thus a reduction of symbiosis during these periods due to a low-light intensity habitat. However, this would not explain the maintained $\delta^{13}$C-size gradient in symbiotic species, nor account for the less dramatic $\delta^{13}$C excursion recorded in the shells of *S. roesnaesensis*. Therefore, we suspect that decline of the $\Delta\delta^{13}$C values of the smallest *A. soldadoensis* and *M. subbotinae* with respect to those of *S. roesnaesensis* during the ETM2 and PETM is an artifact of differential mixing, with subbotinids rare or absent during the hyperthermal event (Figs 4 and 6).

The cause of decreased subbotinid abundances in early Eocene hyperthermals may be at least in part preservational. There are differences in preservation potential between *Acarinina*, *Morozovella*, and *Subbotina* [33, 50, 55], with *A. soldadoensis* and *M. subbotinae* particularly resistant to dissolution [56]. The potential for differential dissolution is supported by frequent observation of cancellate (subbotinid) shell fragments in our samples. However, if preservation alone were responsible for the observed trends in *Subbotina*, relative abundance should decline with an increase in fragmentation and *vice versa*. This is not consistently observed, nor is dissolution so intense as to leave only fragments $<<$150 μm, as evidenced by the continued presence of whole shells (Fig 6; S4 Fig). Fragmentation remained low (<14%) (Fig 6) and CaCO$_3$ high (>85%) [32] throughout the ETM2 and H2 intervals at Site 1209, although increased dissolution is evidenced by a decrease in CaCO$_3$ in ETM2 of ~ 5 wt % [57]. Alternately, warming, deoxygenation, or increased oligotrophy during hyperthermal events could also have led to an exclusion of subbotinids. Our data do not provide support for changing physical conditions at Site 1209, but we cannot exclude the possibility that such environmental shifts could have impacted subbotinids, which tend to be more common at colder, mid-latitude sites [16, 58, 59]. It is also possible that environmental and preservational factors jointly reduced the abundance of subbotinids at Site 1209.

## Adaptive response of photosymbiosis to post-PETM warming

At Shatsky Rise, size-$\delta^{13}$C in *A. soldadoensis* was lower after the PETM than before, but no such reduction between pre and post event size-$\delta^{13}$C data is observed for the ETM2 or H2 intervals. Furthermore, size-$\delta^{13}$C did not return to pre-PETM values over the course of our record (before at least 52.81 Ma) (Figs 3 and 4). Size-dependent sediment mixing (discussed above) cannot account for the post-PETM size-$\delta^{13}$C observed in *A. soldadoensis* because the geochemical artifact of upwardly displaced shells is limited to periods of rapid isotopic excursion [38, 39]. Rather, we argue that the change in size-$\delta^{13}$C observed in *A. soldadoensis* across the PETM may be an indication of an adaptive shift in photosymbiont ecology.

Size-$\delta^{13}$C is a reliable indicator of symbiosis in modern planktic foraminifera [60], but symbiont loss has yet to be recorded in any living planktic taxa, including those exposed to thermal stress to the extent of inhibiting reproduction and survival [61]. When photosymbiont activity is artificially suppressed in a laboratory setting, planktic foraminifera experience increased mortality, shortened survival times, and reduced overall growth [62]. In keeping with these outcomes, at least some inferences of symbiont loss in the fossil record are associated with unidirectional ecological shifts, population decline, and extinction events [14, 17]. By contrast, during early Eocene hyperthermals, *Acarinina* persisted at high relative abundances at Site 1209, apparently thriving relative to asymbiotic *Subbotina* (Fig 6). A change in size-$\delta^{13}$C in *Acarinina* is indicative of a change in ecology across the PETM which was accompanied by a sharp decrease in abundance (Fig 6) relative to *Morozovella*. We argue that this change in ecology was likely a shift in the type or arrangement of symbiont partners altering foraminiferal $\delta^{13}$C signatures [21, 22, 24–26], rather than expulsion of photosymbionts or 'bleaching'.

Both acarininids and morozovellids remain relatively abundant at Site 1209 through ETM2 and H2 (Fig 6), with acarininids remaining the dominant species except in the youngest sample (53.816 Ma). Despite sustained abundances through the early Eocene hyperthermals at Site 1209, morozovellids begin to decline just prior to (Site 1258), during (Site 1051) and after (Site 1263) the J event (~53.3 Ma) in the Atlantic [48]. During the long-term warm EECO, acarininids became highly abundant while morozovellids declined in abundance [16, 36, 37]. Although, transient shifts in size-$\delta^{13}$C in *Acarinina* have been associated with both the J event and the MECO, in neither case was this associated with a decrease in abundance as was the case for the morozovellids or later *Morozovelloides* [14, 48]. Thus, it is possible that the ability of the acarininids to innovate in symbiosis, typified by the response of *A. soldadoensis* to the PETM, steeled the group against environmental perturbations on very long timescales (millions of years).

Thus, our evidence supports that 1) the shift in size-$\delta^{13}$C in *A. soldadoensis* across the PETM was likely adaptive, allowing this group to thrive through subsequent, less intense hyperthermals and other environmental perturbations, and/or that 2) the ETM2 and H2 hyperthermals did not meet a threshold required to drive further changes to symbiosis.

## Conclusions

The $\delta^{13}$C-size gradient of planktic foraminifera shells has generally been used as evidence for the presence or loss of symbionts during warming on geologic timescales. Records of this widely used proxy for foraminiferal symbiosis (size-$\delta^{13}$C) provide no evidence for a major shift in photosymbiont ecology in either *A. soldadoensis* or *M. subbotinae* over the ETM2 or H2 intervals at Shatsky Rise, although evidence for such a shift during the much more severe hyperthermal, the PETM, exists for *A. soldadoensis*. This demonstrates that hyperthermal events are not consistently linked with 'bleaching', even at sites and in species where changes in symbiosis have been identified. Changes in symbiosis, such as that experienced by *A. soldadoensis* across the PETM, may be adaptive, allowing this species and perhaps other acarininids to thrive through subsequent hyperthermals. Alternatively or additionally, some environmental threshold achieved during the PETM and impacting the long-term trajectory of symbiotic relationships in *A. soldadoensis*, may not have been reached during these lesser hyperthermals.

## Supporting information

**S1 Data. Table listing for each sample the ODP sample name, age, species name, test size class, $\delta^{18}$O and $\delta^{13}$C value, and classification assigned here as "pre", "peak", or "post" event as well as the hyperthermal event the sample was associated with.**
(ZIP)

**S2 Data. Images of all samples prior to stable isotope analysis.**
(ZIP)

**S1 Table. Slope and standard error of the size-$\delta^{13}$C relationship for A. soldadoensis, M. subbotinae, and S. roesnaesensis during the pre-event peak event and post-periods of the H2, ETM2, and PETM intervals.**
(XLSX)

**S1 Fig. Representative examples of the complete samples prepared for isotope analyses.** Shown are the smallest (A-C) and largest (D-F) size fraction analyzed at 53.816 Ma, and the smallest (G-I) and largest (J-L) size fraction analyzed at 56.632 Ma. All sample images have been included as a supplementary dataset.
(PDF)

**S2 Fig. $\delta^{13}$C records from *S. roesnaesensis* with size fraction denoted by color, demonstrating some dampening of $\delta^{13}$C excursions in smaller shell.** The H2, ETM2, and PETM intervals are designated with yellow, orange, and brown shading respectively. All size points are shown in each panel, with differing trends across size fractions shown in colored lines. (PDF)

**S3 Fig.** The slope of $\delta^{13}$C relative to size for *A. soldadoensis* (red), *M. subbotinae* (green), and *S. roesnaesensis* (blue) through time at Site 1209 shown as cartoon foraminifera compared with slopes from multiple species at the genus level for *Acarinina* (red), *Morozovella* (green) and *Subbotina* (blue) from ODP Site 577 [19] rescaled to the age model of [41] shown as open circles with error bars. The H2, ETM2, and PETM intervals are designated with yellow, orange, and brown shading respectively. (PDF)

**S4 Fig.**
(PDF)

# Acknowledgments

We thank B Erkkila and M Wint of the Yale Analytical and Stable Isotope Center for help with isotopic analyses and the International Ocean Discovery Program for samples from Site 1209. JS, SD, ET, & PH conceptualized the project; ET & PH provided resources, funding, and supervision; JS & SD carried out investigations; JS & CVD carried out formal analyses; CVD carried out visualization and original draft preparation; CVD, JS, SD, ET, and PH all contributed substantially to the review and editing process.

# Author Contributions

**Conceptualization:** Jack O. Shaw, Simon D'haenens, Ellen Thomas, Pincelli M. Hull.

**Formal analysis:** Catherine V. Davis, Jack O. Shaw.

**Investigation:** Jack O. Shaw, Simon D'haenens.

**Resources:** Ellen Thomas, Pincelli M. Hull.

**Visualization:** Catherine V. Davis.

**Writing – original draft:** Catherine V. Davis.

**Writing – review & editing:** Catherine V. Davis, Jack O. Shaw, Simon D'haenens, Ellen Thomas, Pincelli M. Hull.

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
