## [Decision Letter · Decision Letter 0]

25 Nov 2021

PONE-D-21-33519­­Photosymbiont associations persisted in planktic foraminifera during early Eocene hyperthermals at Shatsky Rise (Pacific Ocean)PLOS ONE

Dear Dr. Davis,

Thank you for submitting your manuscript to PLOS ONE. I have now received two reviews of your work. Both reviewers highlight a number of corrections that need to be made before publication. After careful evaluation, I am recommending major revisions. When revising your manuscript, please consider all issues mentioned in the reviewers' comments carefully: please outline every change made in response to their comments and provide suitable rebuttals for any comments not addressed. Please note that your revised submission may need to be re-reviewed. 

In the revised version of the paper, please pay full attention and discuss in-depth: preservation state of planktic foraminifera shells (point 1 by rev#1); the meaning of d13C signal *versus* varying C reservoirs in ambient seawater (point 2 by rev#1); take into account the re-organization of figures (both reviewers and myself); improve and implement the discussion of specific points (rev#2 and partially rev#1).

We look forward to receiving your revised manuscript.

Kind regards,

Alessandro Incarbona

Academic Editor

PLOS ONE

Journal Requirements:

2. In your Methods section, please provide additional details regarding the studied materials used in your study and ensure you have described the source. For more information regarding PLOS' policy on materials sharing and reporting, see https://journals.plos.org/plosone/s/materials-and-software-sharing#loc-sharing-materials.

"ET recognizes funding by National Science Foundation (NSF) OCE 1536611. PMH, SD, and JOS recognize funding by NSF OCE 1536604 and a Sloan Research Fellowship. JS, SD, ET, & PH conceptualized the project; ET & PH provided resources, funding, and supervision"

"ET recognizes funding by National Science Foundation (NSF) OCE 1536611. PMH, SD, and JOS recognize funding by NSF OCE 1536604 and a Sloan Research Fellowship. "

5. We noted in your submission details that a portion of your manuscript may have been presented or published elsewhere. "Isotope data from the PETM was previously published in Shaw JO, D'Haenens S, Thomas E, Norris RD, Lyman JA, Bornemann A, et al. Photosymbiosis in planktonic foraminifera across the Paleocene–Eocene thermal maximum. Paleobiology. 2021. 1-16.

This published data is cited throughout and used for comparative purposes to extend our dataset."

Reviewers' comments:

Reviewer's Responses to Questions

**Comments to the Author**

1. Is the manuscript technically sound, and do the data support the conclusions?

Reviewer #1: No

Reviewer #2: Yes

2. Has the statistical analysis been performed appropriately and rigorously? 

Reviewer #1: No

Reviewer #2: Yes

3. Have the authors made all data underlying the findings in their manuscript fully available?

Reviewer #1: Yes

Reviewer #2: Yes

4. Is the manuscript presented in an intelligible fashion and written in standard English?

Reviewer #1: No

Reviewer #2: Yes

5. Review Comments to the Author

Reviewer #1: The paper of Davis and colleagues on the Photosymbiont associations persisted in planktic foraminifera during early Eocene hyperthermals at Shatsky Rise (Pacific Ocean) presents a potentially interesting topic. Unfortunately, the manuscript is poorly developed as it stands now, and suffers from a couple of fatal flaws. Consequently, I would suggest to not accept the paper for publication.

(1) As can be seen from the figures 2 and 5 (or 6? Figure numbers in the main text do not correspond to the figure captions), samples are possibly not available from the PETM “peak event”, other than suggested by figures 1, 4, and 5. Why does the figure 3 not include data points from the warm time intervals? Assuming that reliable isotope data would be present on ETM2 and H2, reasoning on the effects on photosymbionts and the d13C of the different planktonic foraminifera shells would be based on only two time intervals. However, in the lines 177 and 178 of manuscript it says that “Planktic foraminifera from ODP Site 1209 are frosty with poor to moderate preservation throughout the sampled interval”, which brings the entire reasoning into doubt. I’m actually surprised by the assessment on the quality of foraminifera shells, because good preservation is reported in the initial ODP reports. May preservation have suffered over time in the core repository?

(2) The Paleocene and Eocene hyperthermals have been shown to be possibly caused by high atmospheric (and by diffusion marine) methane concentration (e.g., Zachos et al. 2005) and high CO2, both of which molecules would have a strong effect on the d13C signal of the foraminifera species discussed here, plus the effect of decreasing pH on foraminifera shell calcification and preservation. Consequently, the d13C signal may not reflect changing photosymbiont activity, but varying C reservoirs in ambient seawater. These effects may be resolved by assessing mass balance of the different reservoirs. However, taken the differential ecological demands and preservation of the shallow and deep dwelling species discussed here, and the poor temporal resolution of the record, this exercise may be difficult.

(3) Statistical analyses should prove significance of any changes in the d13C.

There are many minor shortcomings in the reasoning and understanding of planktonic foraminifera biology and ecology, such as the (mis-) interpretation of “gigantism” in foraminifera; the northern hemisphere glaciation may be considered time interval of global cooling. The red symbols in the benthic stack in the figure 2 do not represent an age model worth to talk about, but rather match -at the best- the age model of Westerhold in some places.

In general, the figures are poorly executed. Organization of the panels in the figures 1 and S3 make comparison of data over different time intervals almost impossible; the figure 1 is rather data documentation and may be moved to the supplementary materials. The x-axis in the figure 3 may be significantly shortened to move the data points closer together and to see any significant (?) changes. It’s a cute idea to present the data points in the figure S4 as little foraminifera icons, but which does not really help to identify varying shell diameter (size legend).

Reviewer #2: This is an interesting contribution to the understanding of the symbiotic loss during warmer geological intervals.

The paper is clearly written but sometimes it is too concise, also I feel the absence of additional explanation especially in the discussion section. In my opinion the discussion needs to be implemented with more detailed explanation and examples and comparison with published data that are only mentioned in the text but not discussed.

There are some issues related to the presentation of the figures and, consequently, to the explanation in the text that need to be carefully considered. I have listed them below:

Line 133-135- In Fig. 2 there is eocaena not roesnaesensis. Check for consistency.

Line 183-184. Not clear I suppose it is Fig. 2 as fig. 1 only shows the d13C record. I do not see that a partial signal is retained associated to benthic hyperthermal because soldadoensis and eocaena yield negative values also in the intervals in between the hyperthermal.

Line 197-202. The text of the caption refers to Fig. 4 not Fig. 3.

Line 198-199. Again, I am confused, which species of Subbotina has been analysed?

Line 213-214. I do not see any blue line for Subbotina in Fig. 5

Line 224-225. There is no Fig. 6 in the copy I have.

Line 264. Fig. 6 is missing.

Line 274- The order of the Supplementary material needs to be revised, Supplemental Fig. 3 cannot be cited in the text before 1 and 2. There is some confusion in the order of the Supplementary material and it is not clear where the supplementary material need to be cited in the text. Please check and revise text and figures accordingly.

Line 283-284. Not clear, Fig. 5 does not show the continued preservation of whole test. I suppose there is something missing here.

Line 257-327. I find these two chapters a little confused and repetitive. I also miss the explanation and the citations in the text of the figures included in the Supplementary material. I would advise to use those figures to support the explanation and hypothesis provided in the discussion.

Another suggestion is to think about a synthesis figure or scheme that summarize the conclusions presented in the text.

6. PLOS authors have the option to publish the peer review history of their article (what does this mean?). If published, this will include your full peer review and any attached files.

Reviewer #1: No

Reviewer #2: No

---

## [Author Response · Author response to Decision Letter 0]

4 Jan 2022

We’d like to thank both reviewers for their critical assessment of our manuscript. We have tried to address each of the issues raised below. 

Reviewer 1

“As can be seen from the figures 2 and 5 (or 6? Figure numbers in the main text do not correspond to the figure captions)…” 

We apologize for this mistake, and this has been corrected throughout.

“samples are possibly not available from the PETM “peak event”, other than suggested by figures 1, 4, and 5.” 

We did not produce data for the PETM. The data for the PETM is from the publication by Shaw et al., (2021), and used as a comparison to our records from the ETM2 and H2 events. We used all data available for the PETM at this site in our comparison.

“Why does the figure 3 not include data points from the warm time intervals?”

Figure 3 presents a reanalysis of data from Shackleton et al. (1985), which we found interesting in that they show the same general trend as our data over an overlapping time period. The original dataset did not sample the lesser hyperthermals.

“Assuming that reliable isotope data would be present on ETM2 and H2, reasoning on the effects on photosymbionts and the d13C of the different planktonic foraminifera shells would be based on only two time intervals.” 

Yes, we have focused on two hyperthermal events and flanking time intervals for our study, for a total of 11 time slices. But we have also included data reanalysis from a third event and flanking intervals (PETM). These are the more prominent hyperthermals. 

“However, in the lines 177 and 178 of manuscript it says that “Planktic foraminifera from ODP Site 1209 are frosty with poor to moderate preservation throughout the sampled interval”, which brings the entire reasoning into doubt. I’m actually surprised by the assessment on the quality of foraminifera shells, because good preservation is reported in the initial ODP reports. May preservation have suffered over time in the core repository?”

Some confusion exists about what 'good preservation' means exactly, and how different authors at different times expressed it, which is due to development of the concept of preservation state of planktic foraminifera during the last few decades. We probably did not explain this clearly for a non-specialist audience. Most samples obtained by DSDP, ODP and IODP contain planktic foraminifera assessed as 'well preserved' in the past, i.e., before studies by e.g. Pearson et al, 2001; Sexton et al., 2006, who documented that there was considerable recrystallization even of these so-called 'well-preserved' foraminifera. Well preserved foraminifera as now defined ('glassy' preservation) are generally absent in carbonate oozes (such as we studied), the most common sediment recovered by ocean drilling for paleoceanographic studies. Glassy preservation occurs only in clay rich sediments, whereas carbonate oozes contain 'frosty' specimens, as the ones we studied. As widely documented, this recrystallization affects oxygen isotope signatures strongly (even in sediments as young as Miocene), but carbon isotope signatures (which we are investigating here) are very little affected, because there is so much less dissolved carbon in pore waters than oxygen in water. This is explained in detail and well documented for the PETM specimens in Shaw et al., 2021, to which we now more clearly refer; the preservation of specimens is also shown in the supplemental data to that paper available at https://doi.pangaea.de/10.1594/PANGAEA.918702. For this manuscript, photographs of specimens showing preservation are shown in supplemental Figure 1 and the Supplemental data. We have also revised our discussion of preservation on lines 170-180 to reflect some of this nuance.

Our assessment follows Petrizzo et al., (2008) in their evaluation of the PETM, as well as the documentation in Shaw et al. (2021). Thus, our statement of ‘poor to moderate’ is a conservative estimate of preservation, as it refers to likely the interval of worst preservation in any considered interval. We have revised to ‘moderate’ to better reflect the description of sediments well above the PETM. Unfortunately, there are not sufficient constraints to determine whether preservation has changed during core storage based on our sample set alone. In general, changes in preservation of planktic foraminifera have not been observed in organic-poor carbonate oozes such as studied here, even those stored for decades, i.e., obtained during the earlier DSDP expeditions in the early 1970s, under the cooled conditions in the core repositories. 

“The Paleocene and Eocene hyperthermals have been shown to be possibly caused by high atmospheric (and by diffusion marine) methane concentration (e.g., Zachos et al. 2005) and high CO2, both of which molecules would have a strong effect on the d13C signal of the foraminifera species discussed here, plus the effect of decreasing pH on foraminifera shell calcification and preservation. Consequently, the d13C signal may not reflect changing photosymbiont activity, but varying C reservoirs in ambient seawater. These effects may be resolved by assessing mass balance of the different reservoirs. However, taken the differential ecological demands and preservation of the shallow and deep dwelling species discussed here, and the poor temporal resolution of the record, this exercise may be difficult.”

We certainly do not discount nor disregard a change in carbon reservoirs over the time period studied, since this change in reservoir values of �13C is indeed a well studied topic. We used the available information to choose our time-slices to explicitly target the δ13C excursions associated with changing global CO2 and carbon cycling to better understand the ecological response of multiple species of foraminifera to those perturbations. We have tried to make this more explicit in the manuscript, to address a broader audience. Changing carbonate chemistry and source would indeed be expected to the drive δ13C of a bulk foraminiferal sample (or a sample within any given size bin) but does not impact the relative difference in δ13C with size within an individual species in an individual time slice. After all, each species within a time slice lives within the same macro-carbon environment. Environmental δ13C values and/or pH thus change between time slices, but different size classes of the same species living at the same time are living consistently in in a δ13C environment typical for its time. The parameter which we use, i.e., the size-δ13C relation, has been widely interpreted as an indication of photosymbiont activity, after ensuring that the studied specimens indeed were living at the same time at the same place, thus seeing the same background δ13C value in dissolved inorganic carbon and the same pH (Spero & Lea, 1993; Berger et al., 1981; Shackleton et al., 1985; D'Hondt et al., 1994; Norris, 1996; Birch et al., 2012; Edgar et al., 2013; Shaw et al., 2021; Si and Aubry, 2018; Wade et al., 2008). The goal of this paper is to better understand ecological response at the species-level to large scale perturbations, rather than characterize the Eocene carbon environment at high resolution. We have attempted to further clarify this on lines 75-77 and 176-180.

“Statistical analyses should prove significance of any changes in the d13C”

Statistical analyses (e.g., ANCOVA) were carried out, and are described in the document and supplemental materials. In most cases these were not found to be significant, which is one of the major findings of this manuscript. If the reviewer could clarify if and where further tests are needed, we’d be happy to revise accordingly.

“There are many minor shortcomings in the reasoning and understanding of planktonic foraminifera biology and ecology, such as the (mis-) interpretation of “gigantism” in foraminifera; the northern hemisphere glaciation may be considered time interval of global cooling.” 

If the reviewer would specify where misunderstandings are found, we would be glad to address them in a future revision. With reference to “gigantism,” this was discussed in summarizing the findings of another paper that used this exact terminology. However, we have removed this reference entirely as the discussion likely strays too far from our main story. 

“The red symbols in the benthic stack in the figure 2 do not represent an age model worth to talk about, but rather match -at the best- the age model of Westerhold in some places.”

We do not present a new age model for this site but use the age model of Westerhold et al. (2018) throughout. Figure 2 includes stable isotope data for multiple species of benthic foraminifera during the relevant time slices at the site, superimposed on the more complete Westerhold et al. record. This is neither a stack, nor a revision of any existing age model, but a comparison of our data with a published record. We have attempted to clarify this in Figure caption 2, since we think that this is a misunderstanding of our figure.

“Organization of the panels in the figures 1 and S3 make comparison of data over different time intervals almost impossible the figure 1 is rather data documentation and may be moved to the supplementary materials.” 

We have revised the presentation of Figure 1 and S3, but respectfully disagree with the reviewer regarding removing Figure 1 from the main text. We believe it is important to present our size-δ13C data in a comparable manner to how it is has historically been shown in similar analyses, for example by Wade et al, (2008) and Edgar et al., (2013). We also think that presenting these data is important for following our argument, by a broader audience which may not be as familiar with these types of data.

“The x-axis in the figure 3 may be significantly shortened to move the data points closer together and to see any significant (?) changes.” 

We have changed the scale on Figure 3 as suggested.

“It’s a cute idea to present the data points in the figure S4 as little foraminifera icons, but which does not really help to identify varying shell diameter (size legend).”

Thank you for this suggestion. This figure has been removed altogether in an effort to streamline our Supplement in response to Reviewer #2. 

Reviewer 2

“In my opinion the discussion needs to be implemented with more detailed explanation and examples and comparison with published data that are only mentioned in the text but not discussed.”

 We have reorganized some the discussion and added additional explanation 

where we saw that this may be useful. If there are additional sections which require elaboration, please let us know. 

“Line 133-135- In Fig. 2 there is eocaena not roesnaesensis. Check for consistency.”

Thank you for catching this. We apologize for this mistake, and it has been corrected throughout

“Line 183-184. Not clear I suppose it is Fig. 2 as fig. 1 only shows the d13C record. I do not see that a partial signal is retained associated to benthic hyperthermal because soldadoensis and eocaena yield negative values also in the intervals in between the hyperthermal.”

Our apologies for the confusion around figure numbers; we made a mistake. This has been corrected throughout and the statement has been removed entirely.

“Line 197-202. The text of the caption refers to Fig. 4 not Fig. 3.”

This has been corrected.

“Line 198-199. Again, I am confused, which species of Subbotina has been analysed?”

S. roesnaesensis was analysed. An earlier draft had included the incorrect species name, but this has been corrected throughout.

“Line 213-214. I do not see any blue line for Subbotina in Fig. 5”

 This line at “0” has been added. 

“Line 224-225. There is no Fig. 6 in the copy I have.” & “Line 264. Fig. 6 is missing.”

 Our apologies if this was a submission error. It should be included in this version.

“Line 274- The order of the Supplementary material needs to be revised, Supplemental Fig. 3 cannot be cited in the text before 1 and 2. There is some confusion in the order of the Supplementary material and it is not clear where the supplementary material need to be cited in the text. Please check and revise text and figures accordingly.”

 The supplementary material has been reordered and cited accordingly.

“Line 283-284. Not clear, Fig. 5 does not show the continued preservation of whole test. I suppose there is something missing here.”

This statement should refer to Figure 6. We have checked and corrected figure references throughout.

“Line 257-327. I find these two chapters a little confused and repetitive. I also miss the explanation and the citations in the text of the figures included in the Supplementary material. I would advise to use those figures to support the explanation and hypothesis provided in the discussion.”

Both sections have been edited and reorganized to improve clarity, remove repetition, and include reference to supplementary figures.

“Another suggestion is to think about a synthesis figure or scheme that summarize the conclusions presented in the text.”

We appreciate this suggestion but have been unable to summarize these ideas more clearly through a simple visualization. It is our hope that the revisions made to discussion and conclusion will improve clarity sufficiently to address this point.

---

## [Decision Letter · Decision Letter 1]

14 Mar 2022

PONE-D-21-33519R1­­Photosymbiont associations persisted in planktic foraminifera during early Eocene hyperthermals at Shatsky Rise (Pacific Ocean)PLOS ONE

Dear Dr. Davis,

Thank you for submitting your manuscript to PLOS ONE. After careful consideration, we feel that it has merit but does not fully meet PLOS ONE’s publication criteria as it currently stands. Therefore, we invite you to submit a revised version of the manuscript that addresses the points raised during the review process.

Three main concerns have been raised by referee#3. I agree that a thorough revision is still needed before publication, and I invite you to fully reply to weaknesses of your argumantation.

We look forward to receiving your revised manuscript.

Kind regards,

Alessandro Incarbona

Academic Editor

PLOS ONE

Reviewers' comments:

Reviewer's Responses to Questions

**Comments to the Author**

1. If the authors have adequately addressed your comments raised in a previous round of review and you feel that this manuscript is now acceptable for publication, you may indicate that here to bypass the “Comments to the Author” section, enter your conflict of interest statement in the “Confidential to Editor” section, and submit your "Accept" recommendation.

Reviewer #3: (No Response)

2. Is the manuscript technically sound, and do the data support the conclusions?

Reviewer #3: No

3. Has the statistical analysis been performed appropriately and rigorously? 

Reviewer #3: N/A

4. Have the authors made all data underlying the findings in their manuscript fully available?

Reviewer #3: Yes

5. Is the manuscript presented in an intelligible fashion and written in standard English?

Reviewer #3: Yes

6. Review Comments to the Author

Reviewer #3: Comment to authors

The paper “Photosymbiont associations persisted in planktic foraminifera during early Eocene hyperthermals at Shatsky Rise” by Davis et al., examine potential planktic foraminiferal bleaching during less intense early Eocene hyperthermals as compared with the most extreme PETM event. The biotic response to climate warming is crucial especially in the context of the ongoing climatic changes. The argument is therefore attractive for PLOS ONE readers.

The authors suppose the maintaining of symbiotic relationship in mixed-layer planktic foraminifera across early Eocene hypertermals.

However, there are several issues that should be clarified to reach the adequate high level expected from a PLOS ONE publication. The most critical issue is the interpretations of data that do not appear sufficient to substantiate the conclusions.

Actually, across the H2 and to a less degree at the ETM2, Acarinina soldadoensis show a reduction of symbiotic activity. The authors interpret this evidence through the ‘upward mixing’ of smaller forms, in agreement to Hupp et al. 2019, Hupp and Kelly 2020 and Shaw et al. 2021.

There are at least three problems with this interpretation in relation with this work. The first is that, also admitting small acarininids may have pushed upward, surely small acarininids were living besides larger forms during the minor hyperthermals. How is it possible to determine the influence of the two groups? To validate their hypothesis, they should have been sure to have analysed only the smaller older tests….

The second problem is that they do not show the abundance of the acarininid species. Hupp et al. 2019 and Hupp and Kelly 2020 studying the PETM at Site 690, support the possible maintaining of symbiosis despite the loss of size-dependent δ13C signatures in Acarinina subsphaerica as this species is more abundant than A. soldadoensis within the pre-CIE interval thus more subject to be brought upward. The latter species records a less evident offset in CIE due to the opposite. Nevertheless, if the large part of small A. subspherica were pushed upward, a decrease in abundance should have recorded just below the CIE shift. Whatever the possible interpretation of abundances, these data are not provided in this paper at the site studied by the authors. Hupp et al. 2019, aware of the problems above, do not discard the bleaching hypothesis though preferring the upward mixing interpretation.

The third problem is to explain why small morozovellis and subbotinids were not brought upward? Also in this case, abundant of the species studied would have helped. The sentence that acarininids and morozovellids are dominant in all intervals (line 211) is not sufficient. Moreover, acarininnids show a decrease in abundance at the PETM, in the upper part of the ETM2 and even more markedly at the H2. What about A. soldadoensis?

These decrease in acarninids abundance and the coupled increase of moroozvellids are not discussed/interpreted.

The evidence that all the d13C values of A. soldadoensis and M. subbotinae during hyperthermals is closer to that from Subbotina, is not adequately examined/discussed. This record can suggest possible A. soldadoensis habitat deepening where light is attenuated, and symbiosis is reduced.

Finally, the hypothesis that the planktic foraminiferal symbiosis relationship changed at the PETM thus allowing acarininids to better tolerate the impact of minor hyperthermals, is interesting but needs further support. As an example, Bornemann & Norris, 2007; Hemleben et al., 1989 show that living planktic foraminifera bearing chrysophyte symbionts have a δ13C–test size gradient much lower than those hosting dinoflagellates. The authors should investigate whether there are literature data on changes in abundance/evolution/migration on microalgae symbionts across the studied interval that could shed light interpretation. The first step can be verifying whether changes abundance/diversity of potential symbiont microalgae a are recorded at the site studied. In addition, the acarininid resilience post-PETM related to different symbionts or change in symbiotic mechanism contrasts with bleaching episodes recorded at the EECO and MECO (see below).

Part of the discussion/interpretation is at present in the results paragraph.

As final points, some of the species shown in supplementary figures may belong to A. coalingensis rather that A. soldadoensis. The firs species (three final chambers rather that four, more globose, compact test) is generally smaller that A. soldadoensis, so it appears that the ‘crucial’ small A. sodadoensis could be A. coalingensis.

Below other comments:

ABSTRACT

Line 36-37: in the present form of the sentence, it is not clear how preservation and decline in abundance of subbotinids have diminished the difference in d13C with symbiotic taxa. Is the preservation of S. roesnaesensis different from M. subbotinae and A. soldadoensis? How to explain this difference? Fragmentation is not high, especially at H1. Yourseff use these low fragmentation values to explain that the low subbotinid abundance at the hyperthermals is not a taphonomic artefact. This record can rather suggest possible A. soldadoensis habitat deepening where light is attenuated, and symbiosis is reduced. (see also above and comment on figure 5).

INTRODUCTION:

Line 56:….unclear, especially regarding planktic foraminifera.

Line 66: Following your hypothesis that PETM changed the acarinid symbiosis allowing their resilience, how can you explain that bleaching episode occurred just after the ‘minor’ hyperthermal J at the EECO (Luciani et al. 2017 Paleoceanogr.) and during the less intense warming at the MECO (Edgar et al.2013, Geology)?

Line 67 add references.

Line 69-70: the �13C enrichment with respect to coexinting asymbiotic taxa derive from the surface water habitat where algae remove preferentially the light isotope, as they do at a greater extent at the increasing test-size. Some Cretaceous taxa showing this enrichment lived in the upper water column but were asymbiotic.

Lines 96-99: Suggest to expand the sentence in order to briefly explain the problems in interpreting the past planktic foraminifera symbiotic relationship.

METHODS:

Add method(s) adopted to evaluate fragmentation/dissolution.

Line 112: Add the temperature used to dry the samples. As you know temperatures >50 °C may affect the results. The use of sodium metaphosphate can modify the isotope analyses?

Line 115: Explain here the rationale behind the choose of the 6 size fractions.

Line 126: indicate which species of benthic forams.

Line 141: not clear in the present form of the text how you analysed 199 foraminiferal samples from the 41 samples cited. Explain that 199 derive from the different test-size per sample (?)

Line 145: explain here or in the introduction the rationale of the size-d13C analyses in relation with photosymbiotic relationship/bleaching.

Line 146: Suggest specifying here from which levels come the [15] data.

Line 161: in the same 41 samples from which isotopes were analyzed.

RESULTS

Here, results of fragmentation should be added also .

Line 176-180: the sentences should be moved in the discussion and better rephrased, especially lines 177-170 that are not clear.

Line 210-211: expand results on planktic foraminiferal abundance, e.g. percentage and variations also for acarininids and morozovellids, not only for subbotinids.

Line 211: see above about the need of abundance of species analysed.

DISCUSSION

This introductive paragraph should be re-organized: I suggest starting with the interpretation of authors data about the supposed upward mixing, then move to the problem of test-size related to symbiont bleaching (see note below Line 322-235).

Line 222: Symbiosis maintained across the ETM2 and H2 (add in Acarinna soldadoensis)

Lines 224: clarify which is the ‘shift’ in symbiont ecology, decrease/loss of symbiotic relationship. As in the present form, the reader understands that algae changed their ecology

Line 223: saw=show Wouldn’t it better ‘record’?

Line 243: not clear what do you mean with ’peak event samples considered toghether’ (can samples be defined as peak event? Which event?

Line 225: rather than ‘observed’, Shaw et al. ‘assume/suppose’

Line 226: which genus?

Line 228: add at Site 1209.

Line 231: You stated above that the largest size-fraction sometimes shows one specimen (which species?) or a few specimens (which species?) This low numbers are not sufficient to say that there was not acarininid test-size reduction.

Line 232: were ETM2 and H2 studied by Kaiho et al. 2006 at Site 1210? This is the first time that Site 1210 is cited, no other significant information from this very close site?

Line 233-235: As noticed for the reliability of isotope data from one/few specimens in largest fraction, few specimens of large size are not enough to say that there was not size decreasing and therefore to support conserving of the symbiotic relationship. More detailed quantitative data on size tests are needed.

RARITY OR ABSENCE OF SUBBOTINA IN HYPERTHERMALS

Line 249: delete the second ‘at Site 1209’

Line 254: There are also data from the Tethys (e.g., Pardo et al., 1999 JFR; Arenillas and Molina, 2000 Revista Española de Micropaleontología; Arenillas et l. 2000 International Journal of Earth Sciences; Luciani et al. 2007 MarMic). Delete ‘at the height’, at the PETM is enough. Data on planktic foram abundance from the PETM are exclusively from 41-44?, not clear.

Line 254: ref 46, D’Onofrio et al. 2020, do not refer to the PETM but to the interval from H1to T events. The decrease of subbotinids is a common feature of early Eocene hyperthermals, also during the EECO (Luciani et al. 2016 ClimPast, 2017, Paleoceanogr, 2017 GloPlaCha, D’Onofrio et al. 2020 Geosciences. In the PETM from Forada section (northern Italy), subbotinids markedly decrease (Luciani et al.2007 MarMicropaleont.)

Line 255: decrease of subbotinids is recorded by Luciani et al. 2017 GloPlaCha at Walvis Ridge at the H2.

256-257: Not clear to me how the rarity of subbotinids may confirm the upward-mixing if they were rare also below the events. Nor clear the meaning here of a downward mixing post event.

Line 257: the mixing should have added small subbotinids, not clear!

As you recognize, the rarity of subbotinids can be related to the carbonate dissolution that is often recorded across the hyperthermasls due to lysocline/CCD rise, as this group is recognized as dissolution-prone with respect to acarininids and morozovellids. You should have presented the dissolution proxies, e.g., fragmentation index, WPCF, P/B not simply cite two values (calculated according to?)

Line 274: explain increased oligotrophy, based on?

Line 274 or end of section: note that the reduced subbotinid abundance can also be related to elevated ocean temperatures that may have induced more efficient recycling of carbon and nutrients higher in the water column due to enhanced bacterial respiration rate and remineralization (e.g., John et al., 2013 Philos. Trans. R. Soc. A Math. Phys. Eng. Sci. ; ., John et al., 2014 Palaeogeogr., Palaeoclimatol. Palaeoecol.; Pearson and Coxall, 2014 Paleontology). This would have contracted the food supply at depth which, together with possible warmer temperatures, might have led to a consequent reduction of the deeper dwelling niche of subbotinids.

Line 282-283 explain.Line 284-285: Post-PETM size what? Explain the rationale of the upward mixing only during the rapid isotope excursions.

Line 285: it is important to better explain here the mechanisms involved in maintaining/losing the symbiotic relationship.

Line 287: Do you mean that at the PETM A. soldadoensis simply changed their photosymbionts but did not lose the symbiotic relationship?

Line 301-Morozovellids decline in abundance ca 20kyr before the J event at Site 1258, exactly at the J event at Site 1051 and ca 165kyr after J at Site 1263.

Line 302 Morozovelloides not Morozovella became extinct at the Bartonian-Priabonian boundary due to a ‘pre-extiction event’, not in relation with the MECO warming.

Line 305: why the PETM was not involved in acarininid upward mixing?

Line 312: this method is also used to detect photosymbiosis.

Line 317: cite ‘bleaching’ and its significant above, e.g., in the Introduction or discussion.

Line 318 sentence not clear, see note on line 287.

FIGURES

Figure 1 Caption: ‘significant’ on which basis? Why data at 55.932 and 53.968 are not significant? Explain. Note the general comments about the decrease in size-d13C values at peak H2 and ETM2 that suggest bleaching. How is shown the standard error? Add that data form PETM are from [33].

Figure 2: H2 data are from only two samples? According to the supposed upward-mixing, the true CIE onset should be given by the record of largest tests, not involved in upward mixing. Move the capital letters (A to H) exactly over the data column. At present they are all shifted on the right.

Figure 3: relative to increasing test size. Not clear the importance of this figures where no data at the hyperthermals are present.

Figure 4: As noted from the general comments, the authors should discuss the isotope values of S. roesnaesensis that become closer to values from A. soldadoensis and M. subbotinae. Add that data form PETM are from [33]. What represent the various squares/circles in B)? What is the meaning of lightly coloured bands in B)?

Figure 5: see comment above about the decrease of delta Acar-Sub and Mor-Sub at the hyperthermals.

The evidence that all the d13C values of A. soldadoensis and M. subbotinae during hyperthermals is closer to that from Subbotina, is not adequately examined/discussed. This record can suggest possible A. soldadoensis habitat deepening where light is attenuated, and symbiosis is reduced.

Figure 6: explain the fragmentation increase below H2; see also note on methods used to estimate fragmentation. No fragmentation data from PETM are shown.

7. PLOS authors have the option to publish the peer review history of their article (what does this mean?). If published, this will include your full peer review and any attached files.

Reviewer #3: No

---

## [Author Response · Author response to Decision Letter 1]

9 Apr 2022

We’d like to thank the reviewer for their continued careful reading of our manuscript, and for suggesting ways in which we can amend and clarify throughout. 

Before addressing each critique point-by-point, we’d like to brief address the concept of size-dependent, or differential, sediment mixing as I think many of the misunderstandings here stem from questions regarding sediment mixing processes. As an overview, we would expect all oxic marine sediments to experience some mixing (via bioturbation, but also potentially winnowing, etc.). This will preferentially mobilize smaller grains upwards. Thus, when picking a population of foraminifera for geochemical analyses, we always expect some individuals in the sample to have been displaced. Due to the effect of grain size sorting, displacement of smaller individuals is going to be more likely. This is shown very elegantly in a PETM context by Hupp et al. 2019, Hupp and Kelly 2020 as well as Hupp & Kelly, 2022 which is now cited. Mixing is a continuous process and in most cases is going to result in an averaging or “smearing” of the geochemical record when based on bulk analyses. This is not a feature specific to any particular core or site but should always be a consideration when working in the geological record.

Although mixing is more-or-less continuous, where we would expect to actually observe a significant effect is around abrupt events – in our case, those would be where the isotope geochemistry of displaced (e.g., upwardly mixed) and in situ foraminifera would be markedly different. In our record this would be during the H2 and ETM2 events. Mixing of sediment from different ages is occurring outside of these events but would not be geochemically evident because the mixing endmembers are not distinct. Similarly, when a species decreases in abundance, we expect a continued upward “smearing” of individuals from these species, bearing with them an older signal, both of species occurrence and geochemically. It is thus in intervals of rapid change (e.g., including a decrease in species abundance) that the consequences of size-dependent mixing are most apparent. We have tried to clarify these concepts for a more general audience throughout. However, this is not the focus of our manuscript, and we therefore should direct both the reviewer and interested readers to the work dedicated to understanding how mixing processes impact foraminiferal isotopic signals over the Eocene hyperthermals (most relevantly Hupp et al. 2019, Hupp and Kelly 2020, and Hupp & Kelly, 2022). 

Several comments by the reviewer relate to the issue of differential bioturbation. We address each point raised below, but we think that the comments can be mainly addressed by our general notes on differential bioturbation and have indicated where this is the case.

“Actually, across the H2 and to a less degree at the ETM2, Acarinina soldadoensis show a reduction of symbiotic activity. The authors interpret this evidence through the ‘upward mixing’ of smaller forms, in agreement to Hupp et al. 2019, Hupp and Kelly 2020 and Shaw et al. 2021.There are at least three problems with this interpretation in relation with this work. The first is that, also admitting small acarininids may have pushed upward, surely small acarininids were living besides larger forms during the minor hyperthermals. How is it possible to determine the influence of the two groups? To validate their hypothesis, they should have been sure to have analysed only the smaller older tests….”

Both small and larger acarininids were analysed separately. We think that indeed both large and small individuals of these taxa were living during hyperthermals, but smaller pre-event shells are more likely to be included in peak-event populations due to the probabilistic effects of size-dependent sediment mixing. 

“The second problem is that they do not show the abundance of the acarininid species. Hupp et al. 2019 and Hupp and Kelly 2020 studying the PETM at Site 690, support the possible maintaining of symbiosis despite the loss of size-dependent δ13C signatures in Acarinina subsphaerica as this species is more abundant than A. soldadoensis within the pre-CIE interval thus more subject to be brought upward. The latter species records a less evident offset in CIE due to the opposite. Nevertheless, if the large part of small A. subspherica were pushed upward, a decrease in abundance should have recorded just below the CIE shift. Whatever the possible interpretation of abundances, these data are not provided in this paper at the site studied by the authors. Hupp et al. 2019, aware of the problems above, do not discard the bleaching hypothesis though preferring the upward mixing interpretation.” 

See our above general note on differential bioturbation. We would not expect a decrease in abundance below the CIE, but a continual upward “smearing” with a preference for smaller shells being reworked. Relative abundance of A. subsphaerica and A. soldadoensis prior to an event would not impact the likelihood of an individual A. soldadoensis being found within a CIE; we picked only A. soldadoensis. Moreover, while we consider differential mixing processes, we still observe no loss of size-dependent δ13C signatures in Acarinina through our studied interval (slopes remain above 0; Fig. 4; Lines 259-263).

“The third problem is to explain why small morozovellis and subbotinids were not brought upward? Also in this case, abundant of the species studied would have helped. The sentence that acarininids and morozovellids are dominant in all intervals (line 211) is not sufficient. Moreover, acarininnids show a decrease in abundance at the PETM, in the upper part of the ETM2 and even more markedly at the H2. What about A. soldadoensis?”

See our general note on differential bioturbation. Preferential upward mixing occurs in smaller specimens of all species, but the effect is most clear in species with the largest differences in relative abundance across the studied interval. It thus impacts morozovellids and subbotinids, but is probably seen most markedly in subbotinids, as discussed on lines 282-289. We added additional abundance results on lines 221-225 and discussed the change in Acarinina abundance during the PETM (discussed by Shaw et al., 2021) on lines 225-226 and 337-339.

“These decrease in acarninids abundance and the coupled increase of moroozvellids are not discussed/interpreted.”

We now discuss relative abundances of these two species on lines 225-226 and 337-339.

“The evidence that all the d13C values of A. soldadoensis and M. subbotinae during hyperthermals is closer to that from Subbotina, is not adequately examined/discussed. This record can suggest possible A. soldadoensis habitat deepening where light is attenuated, and symbiosis is reduced.”

This possibility has now been explicitly articulated in lines 292-296, but it is not highly probable because reduced symbiosis during hyperthermals should be evident in a reduced �13C-size gradient in both species. This is not observed, thus does not account for the relatively muted �13C excursion in S. roesnaesensis. By contrast, all observations can be explained by sediment mixing processes, as now stated on lines 296-299. 

“Finally, the hypothesis that the planktic foraminiferal symbiosis relationship changed at the PETM thus allowing acarininids to better tolerate the impact of minor hyperthermals, is interesting but needs further support. As an example, Bornemann & Norris, 2007; Hemleben et al., 1989 show that living planktic foraminifera bearing chrysophyte symbionts have a δ13C–test size gradient much lower than those hosting dinoflagellates. The authors should investigate whether there are literature data on changes in abundance/evolution/migration on microalgae symbionts across the studied interval that could shed light interpretation. The first step can be verifying whether changes abundance/diversity of potential symbiont microalgae a are recorded at the site studied.” 

We agree with the reviewer that a change in the type of symbiosis is one possible (likely, even!- also see Gaskell & Hull, 2019) explanation –as stated with the suggested references and others in line 73-75, now reemphasized on line 337-339. If the reviewer is suggesting that we examine the sedimentary record or literature for evidence of free-living candidates for possible foraminiferal symbiosis, I’m afraid this is not feasible. Morphological identification of symbiotic algae in even modern foraminifera is not currently possible. In fact, in the one case where free-living individuals of a foraminiferal symbiont have been cultured, they appear morphologically distinct from their symbiotic counterparts (Spero et al., 1987). Even if purported microalgal cells (which lack a skeleton) were identifiable in sediment samples, any attempt to link them to foraminiferal symbiosis would be wildly speculative at this juncture. This really underscores the importance of using foraminiferal geochemistry to attempt to elucidate these relationships.

“Part of the discussion/interpretation is at present in the results paragraph.”

This has been moved.

“As final points, some of the species shown in supplementary figures may belong to A. coalingensis rather that A. soldadoensis. The firs species (three final chambers rather that four, more globose, compact test) is generally smaller that A. soldadoensis, so it appears that the ‘crucial’ small A. sodadoensis could be A. coalingensis.”

We appreciate this close attention to detail. Upon reexamination of these images, we are confident that this is not a persistent issue (i.e., smaller acaraninid individuals have not been consistently misidentified). In our reexamination we were not able to specifically identify any A. coalingensis in our samples, but it is possible that a few individuals in a large dataset like this one may come under taxonomic dispute, especially because small acaraninids are particularly difficult to identify. We have made available the image of every analyzed specimen, allowing for such disputes and reevaluations to occur and to occur transparently. We emphasize that this is more transparent than current standard practice by either taxonomists or geochemists, and we hope that this practice will become more so in the future. 

“Line 36-37: in the present form of the sentence, it is not clear how preservation and decline in abundance of subbotinids have diminished the difference in d13C with symbiotic taxa.” 

We revised the line in the abstract to explicitly include the linkage to sediment mixing, and have also addressed the issue with an expanded explanation for why a decrease in abundance due to either flux or preservation would lead to an increased mixing signal as referenced above and on lines 282-289. If more subbotinids are from pre-excursion intervals (higher �13C), the excursion as observed in other taxa will appear to diminish the difference in �13C between the two species.

“Is the preservation of S. roesnaesensis different from M. subbotinae and A. soldadoensis? How to explain this difference? Fragmentation is not high, especially at H1. Yourseff use these low fragmentation values to explain that the low subbotinid abundance at the hyperthermals is not a taphonomic artefact. This record can rather suggest possible A. soldadoensis habitat deepening where light is attenuated, and symbiosis is reduced. (see also above and comment on figure 5).”

Yes, it has been well documented that Subbotinids are less dissolution resistant than M. subbotinae and A. soldadoensis (see line 301-303 and referenced literature). We agree that preservation is likely not a major issue at this site: fragmentation is not high, and the decrease in CaCO3 % is not very pronounced at this site (in contrast to other sites, e.g., at greater paleodepth) (see line 310). However, we can’t entirely rule out preservation as a possible driver of low subbotinid abundance, but we suggest that it is unlikely to be the sole driver (304-308). 

“Line 56:….unclear, especially regarding planktic foraminifera.”

We do not think a change is necessary; planktic foraminifera are emerging as a model organism from this time period due to their excellent fossil record. 

“Line 66: Following your hypothesis that PETM changed the acarinid symbiosis allowing their resilience, how can you explain that bleaching episode occurred just after the ‘minor’ hyperthermal J at the EECO (Luciani et al. 2017 Paleoceanogr.) and during the less intense warming at the MECO (Edgar et al.2013, Geology)?”

We think that our text was not fully understood, and have addressed this in part by revising the text and adding references. We did not mean to propose that a single shift in photosymbiosis protected the group from any future symbiotic shifts, but state that ‘it is possible that the ability of the acarininids to innovate in symbiosis, typified by the response of A. soldadoensis to the PETM, steeled the group against environmental perturbations on very long timescales’. If anything, we are arguing that the ability to change symbiosis and persist as a successful clade *is* resilience. In our opinion, potential bleaching of planktic foraminifera during warm periods of the past is a complex issue, and the statement 'bleaching episode occurred' does not fully address worldwide occurrence of this process.

“Line 67 add references.”

References have been added in line. 

“Line 69-70: the �13C enrichment with respect to coexinting asymbiotic taxa derive from the surface water habitat where algae remove preferentially the light isotope, as they do at a greater extent at the increasing test-size. Some Cretaceous taxa showing this enrichment lived in the upper water column but were asymbiotic.”

We fully acknowledge that �13C enrichment alone does not signify symbiosis and have thus restated as “Photosymbiosis in modern and fossil foraminifera can be identified primarily by (1) a positive relationship between shell size and δ13C (size-δ13C), and secondarily by (2) δ13C enrichment of the whole shell with respect to coexisting asymbiotic taxa”. However, this metric is frequently used in concert with the more diagnostic size-δ13C, so we think it is important to state here. 

“Lines 96-99: Suggest to expand the sentence in order to briefly explain the problems in interpreting the past planktic foraminifera symbiotic relationship.”

This has been done, with lines now reading “However, it is difficult to clearly distinguish changes in photosymbiont status across past warming events due to multiple confounding factors. These factors include size dependent sediment mixing, with preferential upward mixing of smaller individuals into younger sediments artificially reducing size-δ13C during isotopic excursions [38,39]. Additionally, changes in light levels and/or foraminiferal depth habitats could have decreased photosymbiont activity, thus reducing both the size-δ13C correlation and the differences in δ13C between symbiotic and asymbiotic taxa [35]. Finally, spatial and taxonomic heterogeneity in the response of planktic foraminifera to early Eocene warming [34,35,40] demonstrates that responses to environmental perturbations were at least in some cases regional and species-specific.”

“Add method(s) adopted to evaluate fragmentation/dissolution.”

Fragmentation methods are articulated on line 175-178. We did not assess dissolution independent of this.

“Line 112: Add the temperature used to dry the samples. As you know temperatures >50 °C may affect the results. The use of sodium metaphosphate can modify the isotope analyses?”

The drying temperature of 40C was added here. The use of sodium metaphosphate is standard practice in foraminiferal analyses (including isotope work) and has never been shown to alter isotopic signatures, though tested in many laboratories. There is also no carbon present in sodium metaphosphate, so it could not alter the �13C signatures. 

“Line 115: Explain here the rationale behind the choose of the 6 size fractions.”

These are the standard sieve fractions used for foraminiferal (and other sediment) work, as clarified in the text. They were the size classes in which sufficient individuals were present and use of standard sieve sizes allows for reproducibility across laboratories with access to traditional picking equipment. Note the use of the same size classes by a wide range of other workers including (but not limited to) Shackleton et al. and Hupp et al., 2019, as well as Edgar et al., 2013, Luciana et al., 2017, and Wade, 2004. 

“Line 126: indicate which species of benthic forams.”

This has been added.

“Line 141: not clear in the present form of the text how you analysed 199 foraminiferal samples from the 41 samples cited. Explain that 199 derive from the different test-size per sample (?)”

We analyzed 3 species from 11 sediment intervals and considered up to 6 size-fractions per species per interval, with some replicates (see the Supplemental Data for a full recounting). We have rephased this as “A total of 199 size-fraction and species-specific samples” and moved up out reference to the supplementary data in which information on each run is available. 

“Line 145: explain here or in the introduction the rationale of the size-d13C analyses in relation with photosymbiotic relationship/bleaching.”

This is discussed in the revised introductory paragraph on lines 68-80.

“Line 146: Suggest specifying here from which levels come the [15] data.”

These have been added on lines 159-160.

“Line 161: in the same 41 samples from which isotopes were analyzed.”

We have added this as “…in the same 11 samples…” as only 11 intervals are analyzed here. 

“Here, results of fragmentation should be added also.”

They should! We have added these results on line 229-231.

“Line 176-180: the sentences should be moved in the discussion and better rephrased, especially lines 177-170 that are not clear.”

This has been moved and rewritten for clarity. 

“Line 210-211: expand results on planktic foraminiferal abundance, e.g. percentage and variations also for acarininids and morozovellids, not only for subbotinids.”

These have been added on now line 221-223.

“Line 211: see above about the need of abundance of species analysed.”

We understand from this and other comments that the reviewer would have liked for us to do complete assemblage work in our samples. However, this was not the focus of our study, and would not bring us closer to answering our key questions about photosymbiont ecology through this period (as addressed specifically in the above comments). Rather, we would point the reviewer to the work of Petrizzo (2007), who already carried out detailed foraminiferal assemblage counts over the relevant intervals at Site 1209. We have added additional reference to this work in the text of our manuscript. 

“This introductive paragraph should be re-organized: I suggest starting with the interpretation of authors data about the supposed upward mixing, then move to the problem of test-size related to symbiont bleaching (see note below Line 322-235).”

We do not agree with this suggestion, because we think it is important to discuss key results first. The consideration of mixing is important in assessing sedimentary samples, but here it is a secondary consideration. With respect to A. soldadoensis it is a relatively minor piece of our interpretation. Mixing is discussed more fully regarding the subbotinids where it is most relevant. 

“Line 222: Symbiosis maintained across the ETM2 and H2 (add in Acarinna soldadoensis)”

Both A. soldadoensis and M. subbotinae have been added to the heading title. 

“Lines 224: clarify which is the ‘shift’ in symbiont ecology, decrease/loss of symbiotic relationship. As in the present form, the reader understands that algae changed their ecology”

This has been amended as “A shift in symbiont ecology (type, arrangement or capacity for symbionts), or…”

“Line 223: saw=show Wouldn’t it better ‘record’?”

This has been rephrased as “In contrast, ETM2 SST records show…”

“Line 243: not clear what do you mean with ’peak event samples considered toghether’ (can samples be defined as peak event? Which event?”

This has been rephrased as “However, when “peak” samples from all three hyperthermals (see methods)….” . Please refer to the materials and methods, and supplemental data (Figs 2, S4, and S1 Data) for further identification of which samples are defined as “peak.” 

“Line 225: rather than ‘observed’, Shaw et al. ‘assume/suppose’”

This has been changed to “inferred”

“Line 226: which genus?”

We specified that these are Acaraninids.

“Line 228: add at Site 1209.”

This has been added.

“Line 231: You stated above that the largest size-fraction sometimes shows one specimen (which species?) or a few specimens (which species?) This low numbers are not sufficient to say that there was not acarininid test-size reduction.”

And

“Line 233-235: As noticed for the reliability of isotope data from one/few specimens in largest fraction, few specimens of large size are not enough to say that there was not size decreasing and therefore to support conserving of the symbiotic relationship. More detailed quantitative data on size tests are needed.”

We have clarified that this can only indicate a maintained maximum size of >355 μm. We do not quantitatively examine species-specific trends in size, but note that the inclusion of one individual in the largest fraction for isotope analyses in most cases is an artifact of the requirements of those analyses (to meet mass requirements), and does not indicate that only 1 individual in that size class was present in the sample. We’ve tried to clarify this on lines 144-145. In addition, the S1 Data includes a full recounting of the number of individuals included in geochemical analyses.

“Line 232: were ETM2 and H2 studied by Kaiho et al. 2006 at Site 1210? This is the first time that Site 1210 is cited, no other significant information from this very close site?”

We have restructured this sentence to clarify that the Kaiho reference should only refer to the PETM (not ETM2 or H2, the focus of this study). 

“Line 249: delete the second ‘at Site 1209’”

Thank you for catching this redundancy; it has been removed.

“Line 254: There are also data from the Tethys (e.g., Pardo et al., 1999 JFR; Arenillas and Molina, 2000 Revista Española de Micropaleontología; Arenillas et l. 2000 International Journal of Earth Sciences; Luciani et al. 2007 MarMic). Delete ‘at the height’, at the PETM is enough. Data on planktic foram abundance from the PETM are exclusively from 41-44?, not clear.”

This line is meant to focus on available data from 1209 specifically; we have amended it to clarify, and added references. 

“Line 254: ref 46, D’Onofrio et al. 2020, do not refer to the PETM but to the interval from H1to T events. The decrease of subbotinids is a common feature of early Eocene hyperthermals, also during the EECO (Luciani et al. 2016 ClimPast, 2017, Paleoceanogr, 2017 GloPlaCha, D’Onofrio et al. 2020 Geosciences. In the PETM from Forada section (northern Italy), subbotinids markedly decrease (Luciani et al.2007 MarMicropaleont.)” and “Line 255: decrease of subbotinids is recorded by Luciani et al. 2017 GloPlaCha at Walvis Ridge at the H2.” 

We have rewritten these sentences and added references. 

256-257: Not clear to me how the rarity of subbotinids may confirm the upward-mixing if they were rare also below the events. 

This has been discussed in our text on differential bioturbation: it is the decrease in abundance that matters, regardless of how abundant the group was prior. With fewer contemporary individuals the likelihood of selecting reworked individuals for analysis increases; if there were no individuals alive during the event, all analyzed specimens would be reworked (see Hupp et al., 2022). 

“Nor clear the meaning here of a downward mixing post event.”

This has been removed.

Line 257: the mixing should have added small subbotinids, not clear!

That is correct, as we explained in our discussion of differential mixing, and a well-known phenomenon. Smaller shells are more likely to be mixed, but mixing is not exclusive to small grain sizes. 

“As you recognize, the rarity of subbotinids can be related to the carbonate dissolution that is often recorded across the hyperthermasls due to lysocline/CCD rise, as this group is recognized as dissolution-prone with respect to acarininids and morozovellids. You should have presented the dissolution proxies, e.g., fragmentation index, WPCF, P/B not simply cite two values (calculated according to?)”

We have attempted to clarify this by directing the reader to Fig. 6 showing fragmentation and to the relevant publications reporting CaCO3 in line rather than at the end of the sentence. 

“Line 274: explain increased oligotrophy, based on?”

We present no new evidence of oligotrophy, but simply state a few potential hypotheses of the changing conditions that could exclude subbotinids. We have tried to clarify as, “Alternately, warming, deoxygenation, or increased oligotrophy during hyperthermal events could have led to an exclusion of subbotinids. Our data do not provide support for changing physical conditions at Site 1209, but we cannot exclude the possibility that such environmental shifts could have impacted subbotinids, which tend to be more common at colder, mid-latitude sites [16,58,59]” Please note that we referred to the literature on this topic as cited on line 254. 

“Line 274 or end of section: note that the reduced subbotinid abundance can also be related to elevated ocean temperatures that may have induced more efficient recycling of carbon and nutrients higher in the water column due to enhanced bacterial respiration rate and remineralization (e.g., John et al., 2013 Philos. Trans. R. Soc. A Math. Phys. Eng. Sci. ; ., John et al., 2014 Palaeogeogr., Palaeoclimatol. Palaeoecol.; Pearson and Coxall, 2014 Paleontology). This would have contracted the food supply at depth which, together with possible warmer temperatures, might have led to a consequent reduction of the deeper dwelling niche of subbotinids.”

This is a really interesting suggestion. This could have played a role in subbotinid abundance through the Early Eocene, but the above cited literature does not make clear whether such a contraction would have been associated with hyperthermals specifically. Griffith et al. (2021, Bentho-pelagic Decoupling: The Marine Biological Carbon Pump During Eocene Hyperthermals. Paleoceanography and Paleoclimatology, 36, e2020PA004053) present evidence that this was indeed the case during the PETM, ETM2 and ETM3 at Walvis Ridge Site 1263, but data presented in an abstract (Griffith, E. M., Thomas, E., Lewis, A. R., Westerhold, T., and Winguth, A. M. E., 2019. Reconstructing the marine biological pump across Eocene hyperthermals at Shatsky Rise Site 1209. 13th International Conference on Paleoceanography, September 2-6, Sydney, Australia; Poster Session 1.) suggest that this may not have been so at Shatsky Rise sites. Therefore, we think that a case could certainly be made with the appropriate data (especially because subbotinids do not appear to decline significantly at eutrophic site 1051) but we feel that to do so in this context would be overly speculative. It would be outside the range of this work, which is focusing on symbiont-bearing taxa, not subbotinid ecological niches. 

“Line 282-283 explain.Line 284-285: Post-PETM size what? Explain the rationale of the upward mixing only during the rapid isotope excursions.”

We think this is mainly explained in our general discussion of differential mixing: isotopic signals of mixing would only be apparent if there were changes in isotopic values over a short sedimentary (i.e., time) interval, which we’ve tried to make explicit in the text on lines 322-325. 

“Line 285: it is important to better explain here the mechanisms involved in maintaining/losing the symbiotic relationship.”

We are a little unclear on which the reviewer is asking. The loss of symbiosis in planktic foraminifera has not been observed in any way other than inferred from isotopic evidence, thus the mechanism is in fact fairly unexplained (see our references to literature, including Gaskell & Hull, 2019). Our data cannot shed further light on this topic beyond what is already stated. 

“Line 287: Do you mean that at the PETM A. soldadoensis simply changed their photosymbionts but did not lose the symbiotic relationship?”

That is one possibility, now stated more clearly on line 337-339. We think that our evidence points to a change in the type, arrangement, or capacity for symbiosis. Unfortunately, we do not have sufficient data to confidently distinguish between these possibilities. 

“Line 301-Morozovellids decline in abundance ca 20kyr before the J event at Site 1258, exactly at the J event at Site 1051 and ca 165kyr after J at Site 1263.”

This has been elaborated upon on lines 243-244. 

“Line 302 Morozovelloides not Morozovella became extinct at the Bartonian-Priabonian boundary due to a ‘pre-extiction event’, not in relation with the MECO warming.”

Thank you for catching our typo. It has been corrected and the sentence rephrased.

“Line 305: why the PETM was not involved in acarininid upward mixing?”

This again touches on the topic of differential mixing; there was such mixing across the PETM, as discussed by Shaw et al. (2021) to whom we refer. 

“Line 312: this method is also used to detect photosymbiosis.”

This has been stated.

“Line 317: cite ‘bleaching’ and its significant above, e.g., in the Introduction or discussion.”

“Bleaching” is discussed in the introduction. 

“Line 318 sentence not clear, see note on line 287.”

We hope that this line has now been clarified in our addressing of previous comments.

Figure 1 Caption: ‘significant’ on which basis? Why data at 55.932 and 53.968 are not significant? Explain. …. How is shown the standard error? Add that data form PETM are from [33].

Figure 1 caption has been rewritten to clarify as “Fig 1. δ13C relative to size class for each interval analyzed. Intervals designated as ‘peak’ events are shaded in yellow, orange, and brown for H2, ETM2, and the PETM events. Intervals proceeding and following “peak” intervals represent those considered “pre” and “post” event samples, respectively. Significant (p-value < 0.05) linear fits are shown in each plot with a standard error envelope at the 95% confidence interval; data for which the relationship between size and δ13C was not statistically significant is shown only as data point. PETM data is from [33].” 

“Figure 2: H2 data are from only two samples? According to the supposed upward-mixing, the true CIE onset should be given by the record of largest tests, not involved in upward mixing. Move the capital letters (A to H) exactly over the data column. At present they are all shifted on the right.”

Yes, there are only two samples from within the peak H2 event. And, yes, such a “lag” was demonstrated at Site 690 and could theoretically be present in 1209, though much higher resolution sampling over the transition to hyperthermals would be required to assess this. However, the chronology of events is adopted from Westerhold et al. as stated in the figure caption, not from our size-specific samples. 

Alpha labels have been moved left.

“Figure 3: relative to increasing test size. Not clear the importance of this figures where no data at the hyperthermals are present.”

Our hypothesis addresses the long-term trend in size-�13C in addition to what occurs during hyperthermals, thus we would prefer to retain this figure, which we consider informative to the reader in placing our data in a longer-term context. 

“Figure 4: As noted from the general comments, the authors should discuss the isotope values of S. roesnaesensis that become closer to values from A. soldadoensis and M. subbotinae. Add that data form PETM are from [33]. What represent the various squares/circles in B)? What is the meaning of lightly coloured bands in B)?”

This is discussed on lines 282-299. We have added the reference and explained the use of colors and symbols in the caption.

“Figure 6: explain the fragmentation increase below H2; see also note on methods used to estimate fragmentation. No fragmentation data from PETM are shown.”

We observe 12% fragmentation below H2, compared to 9 and 8% in flanking samples. We are not prepared to argue that this is a significant change and note that it is well within the non-hyperthermal scatter shown in previous works (~5%-20% shown in Figure 5B of [44]). All PETM data is from [44] and no fragmentation data was available, although it is presented in Figure 5B of that publication, which we have cited as comparison. No new PETM analyses (including fragmentation) are presented here.

---

## [Editor Report · Decision Letter 2]

13 Apr 2022

­­Photosymbiont associations persisted in planktic foraminifera during early Eocene hyperthermals at Shatsky Rise (Pacific Ocean)

PONE-D-21-33519R2

Dear Dr Davis,

After careful evaluation, I am pleased to accept your manuscript, "hotosymbiont associations persisted in planktic foraminifera during early Eocene hyperthermals at Shatsky Rise (Pacific Ocean)", PONE-D-21-33519R2, for publication in the journal. I acknowledge that a thorough effort has been made to address concerns by the three reviewers and I think that the paper is ready to be published. Congratulations!

Please, pay full attention to the following issues, that can be provided in the proof stage: 1) I see just one supplementary data file (images) and not two files as written in the text; 2) figure 6, correct typo ‘onset’ and I do not see a good color contrast between the brown box and the pink curve (Acarinina); 3) check acronyms written in full (e.g. CIE and EECO).

Kind regards,

Alessandro Incarbona

Academic Editor

PLOS ONE

---

## [Editor Report · Acceptance letter]

7 Jun 2022

PONE-D-21-33519R2 

­­Photosymbiont associations persisted in planktic foraminifera during early Eocene hyperthermals at Shatsky Rise (Pacific Ocean) 

Dear Dr. Davis:

I'm pleased to inform you that your manuscript has been deemed suitable for publication in PLOS ONE. Congratulations! Your manuscript is now with our production department. 

Kind regards, 

on behalf of

Professor Alessandro Incarbona 

Academic Editor

PLOS ONE